# Benchmarking integration of single-cell differential expression

Hai C. T. Nguyen [1,6], Bukyung Baik[1,6], Sora Yoon[1,5], Taesung Park[2,3] & Dougu Nam [1,4] ✉

Integration of single-cell RNA sequencing data between different samples has been a major challenge for analyzing cell populations. However, strategies to integrate differential expression analysis of single-cell data remain under-investigated. Here, we benchmark 46 workflows for differential expression analysis of single-cell data with multiple batches. We show that batch effects, sequencing depth and data sparsity substantially impact their performances. Notably, we find that the use of batch-corrected data rarely improves the analysis for sparse data, whereas batch covariate modeling improves the analysis for substantial batch effects. We show that for low depth data, single-cell techniques based on zero-inflation model deteriorate the performance, whereas the analysis of uncorrected data using limmatrend, Wilcoxon test and fixed effects model performs well. We suggest several high-performance methods under different conditions based on various simulation and real data analyses. Additionally, we demonstrate that differential expression analysis for a specific cell type outperforms that of large-scale bulk sample data in prioritizing disease-related genes.

Recent advances in single-cell RNA sequencing (scRNA-seq) techniques have tremendously increased our understanding of cell types and progresses in disease[1,2]. While thousands of cells have been sequenced in individual studies (or samples), integration of scRNA-seq data has been confounded by technical variations between studies, called *batch effects*. In particular, the lack of starting materials in scRNA-seq resulted in highly sparse and noisy data, posing a great challenge to batch-effect correction (BEC) of scRNA-seq data[3,4]. Various BEC algorithms have been developed to accurately discriminate cell types from multiple scRNA-seq datasets[3]. However, the impact of batch effects on gene-based analysis such as differential expression (DE) analysis and the strategies to integrate DE analysis for scRNA-seq data remained underinvestigated. Accurate DE analysis in each cell type across samples (or patients) is instrumental in finding dysregulated genes and functions in disease.

Tran and colleagues[3] recently benchmarked 14 BEC methods for scRNA-seq data, and recommended several high performance methods. Most of the BEC methods exploited the low dimensionality of data and removed the technical differences between matched cells using deep learning or statistical models. Some methods then returned batch-effect-corrected data in the original high dimension (dubbed BEC data) for downstream analysis, whereas others provided only the low dimensional embeddings for efficient annotation of cells. In particular, they tested the use of BEC data for DE analysis (bimod method[5]) under a simple batch condition, where the analysis of BEC data showed a superior performance compared to that of uncorrected data. In contrast, it was suggested that batch alignment could severely distort the high-dimensional observation of genes, making gene-based analysis problematic[6], and DE testing for measured data with technical covariates included in the model was recommended over using BEC data[7].

[1]Department of Biological Sciences, Ulsan National Institute of Science and Technology, Ulsan 44919, Republic of Korea. [2]Department of Statistics, Seoul National University, Seoul 08826, Republic of Korea. [3]Interdisciplinary Program in Bioinformatics, Seoul National University, Seoul 08826, Republic of Korea. [4]Department of Mathematical Sciences, Ulsan National Institute of Science and Technology, Ulsan 44919, Republic of Korea. [5]Present address: Department of Genetics, University of Pennsylvania Perelman School of Medicine, Philadelphia, PA 19104, USA. [6]These authors contributed equally: Hai C. T. Nguyen, Bukyung Baik. ✉e-mail: dougnam@unist.ac.kr

While the BEC methods have been used to reduce or eliminate the technical differences between matched cells, they also introduced artifacts derived from data transformation and estimation of batch differences. Therefore, the possible improvements in DE analysis by using BEC data should be investigated extensively using various DE methods and experimental conditions. In contrast, a statistical model with a batch covariate, denoted as *a covariate model*, used the uncorrected data in each batch when estimating the model parameters with which DE was tested (see Methods)[8–10]. Another possible approach for integrating DE analysis of scRNA-seq data was the meta-analysis, where DE analysis was performed for each batch and the resulting statistics or *p*-values were combined for each gene[11,12].

In this study, we compared various workflows for DE analysis of scRNA-seq data with multiple batches in three different approaches: (1) DE analysis of BEC data, (2) covariate modeling, and (3) meta-analysis. These approaches were referred to *integrative strategies* as opposed to the DE analysis of pooled uncorrected data, denoted as *naïve DE analysis*. We considered "balanced" study design where each batch contained both the sample conditions to be compared, which enabled to accommodate batch effects into DE analysis (Fig. 1). This experimental design has been commonly observed in large-scale single-cell studies where each batch included multiple individuals with various group factors, such as severity of disease, sex, age, ethnic group and clinical status[13,14], or in cancer studies where both tumor and nontumor samples were used from the same patients[2,15]. For unbalanced design, the batch effects were just ignored in DE analysis. See Supplementary Notes for additional explanation on our study design. We used both model-based and model-free simulations of scRNA-seq data, and analyzed the impacts of batch-effects, sequencing depth and data sparsity. Furthermore, we compared the signs and fold changes (FCs) of DE genes before and after BEC to analyze the extent of data distortion.

We analyzed real scRNA-seq data for seven patients with lung adenocarcinoma (LUAD)[15]. Notably, the analysis of LUAD epithelial cells prioritized both known disease genes and prognostic genes significantly better than that of large-scale bulk sample data, demonstrating the high resolution and efficacy of DE analysis of scRNA-seq data (denoted as *scRNA-seq DE analysis*). Finally, we benchmarked DE analysis of large-scale scRNA-seq data for COVID-19 patients[14].

## Results

In total, we benchmarked 46 combinations between ten BEC methods (ZINB-WaVE[16], MNN[17], scMerge[18], Seurat v3[19], limma_BEC[10], scVI[20], scGen[21], Scanorama[22], RISC[23] and ComBat[24]), covariate models, three meta-analysis methods (weighted Fisher (wFisher)[12], fixed effects

model (FEM)[11] and random effects model (REM)[11]), observation weights of ZINB-WaVE[25], pseudobulk data[26] and seven DE methods (DESeq2[9], edgeR[27], edgeR_DetRate[28], limmavoom[10], limmatrend[29], MAST[30] and Wilcoxon test). These combinations are denoted as *DE workflows* in this article. We note that all the ten BEC methods tested here yielded BEC data to be used for DE analysis. See Supplementary Notes on how each DE workflow was implemented. We focused on the comparison between two cell groups (case *vs.* control groups) and tested two and seven batches. For each DE workflow, a threshold of *q*-value <0.05 (Benjamini-Hochberg correction[31]) was used to select differentially expressed genes (DE genes). For simulated data, F-score and area under precision-recall curve (AUPR) were compared between DE workflows. In particular, we used $F_{0.5}$-scores and partial AUPR (denoted as pAUPR) for recall rates <0.5, both of which weighed precision higher than recall; precision has been of particular importance because we often needed to identify a small number of marker genes from sparse and noisy scRNA-seq data. Further justification of using these measures is shown in Supplementary Fig. 1. For real scRNA-seq data, the ranks of known disease genes and prognostic genes, false-positive rates (*p*-value <0.05) and false discoveries (*q*-value <0.05) were compared. Throughout this study, we filtered sparsely expressed genes (zero rate > 0.95), considering that genes rarely expressed in a cell type were less likely to have a substantial role in disease.

### Model-based simulation tests

ScRNA-seq count data were simulated on the basis of negative binomial (NB) model using splatter R package[32]. Sparse data with a high overall zero rate (> 80%) after the gene filtering were simulated for each batch. The batch and group factors were estimated using the principal variance component analysis (PVCA)[33]. We first used a moderate level depth (default; average nonzero count of 77 after gene filtering, denoted as depth-77) and simulated 20% DE genes (10% up and 10% down). The $F_{0.5}$-scores and precision-recall results for two batches were shown in Fig. 2. The experiment was performed for six combinations of "dropout" parameter values and case-control ratios (see Methods), and the resulting $F_{0.5}$-scores and pAUPRs were represented as boxplots and averaged curves, respectively. We tested for both small and large batch effects. In both cases, parametric methods based on MAST, DESeq2, edgeR and limmatrend showed good $F_{0.5}$-scores and pAUPRs. Wilcoxon test applied to log-normalized uncorrected data (denoted as Raw_Wilcox) has been the most widely used for scRNA-seq DE analysis[26]; however, its performance was relatively low for moderate depths. ZINB-WaVE (in short, ZW) provided the observation weights (i.e., dropout probability) that were used to unlock bulk RNA-seq tools to analyze single-cell data[25]. These weights

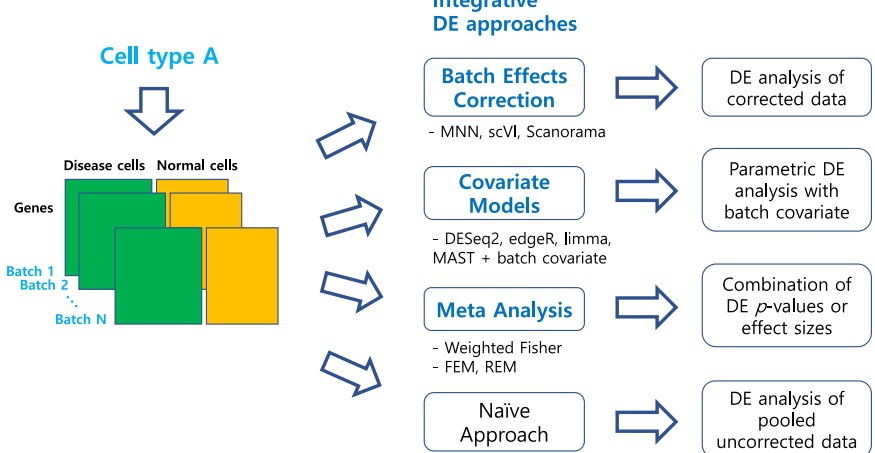

**Fig. 1 | An overview of our benchmark study for differential expression (DE) analysis of scRNA-seq data with multiple batches.** In total, 46 workflows from three integrative strategies and the naïve approach were tested.

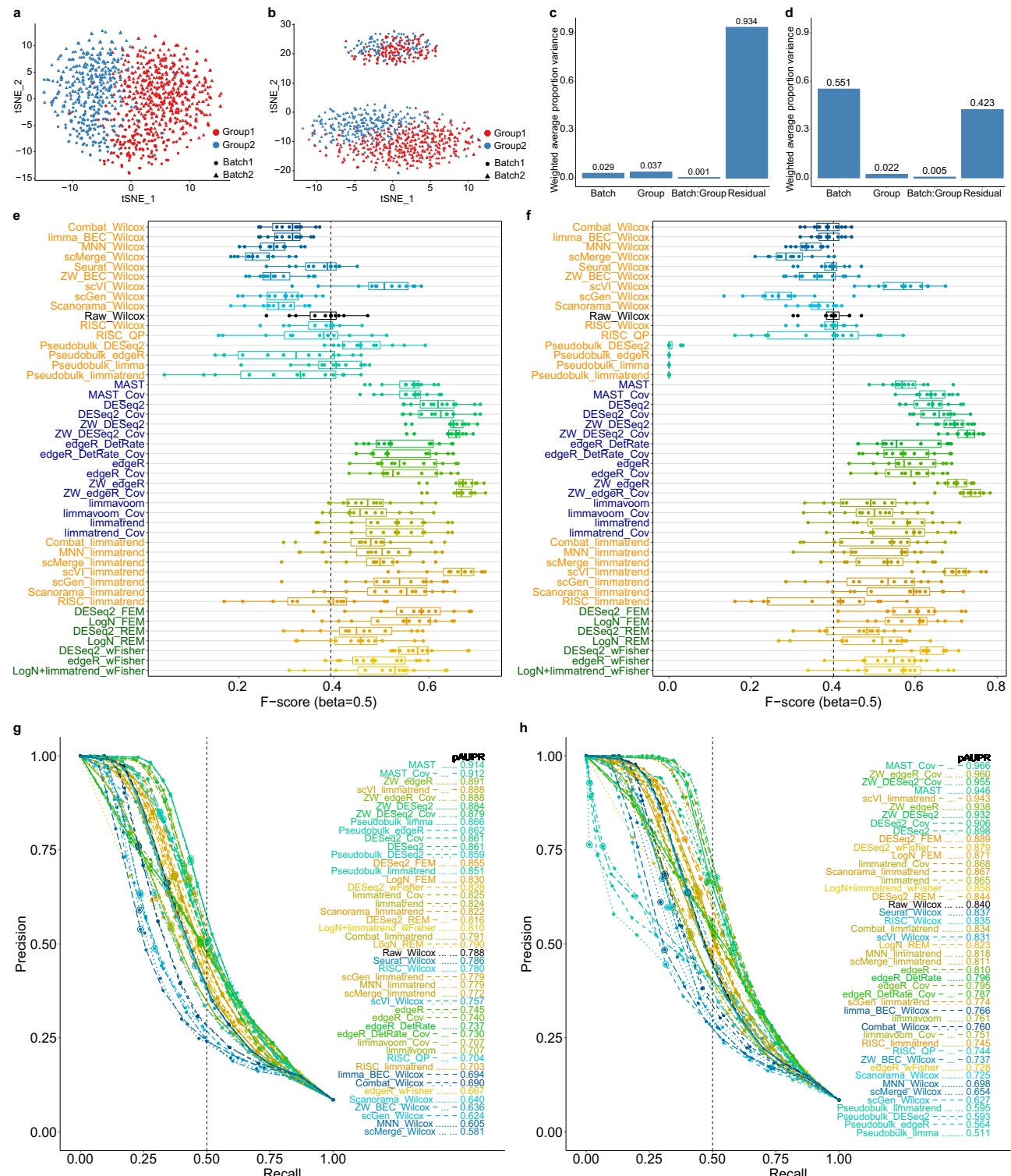

**Fig. 2 | Model-based simulation results for moderate depths (two batches; zero rate >80%).** Scatter plots (tSNE) of two batches for **a** small and **b** large batch effects. Principal variance component analysis results representing **c** small and **d** large batch effects. $F_{0.5}$-scores for 46 differential expression (DE) workflows for **e** small and **f** large batch effects. Results for six cell proportion scenarios (12 instances in total: six for upregulated genes and six for downregulated genes) are represented as boxplots; the lower, center and upper bars represent the 25th, 50th and 75th percentiles, respectively, and the whiskers represent ± 1.5 × interquartile range. The vertical dotted lines (black) indicate the median $F_{0.5}$-score of Wilcoxon test (Raw_Wilcox). Precision-recall curves for **g** small and **h** large batch effects. The partial areas under the curve for recall rate <0.5 (pAUPRs) are computed and sorted in descending order in the legends. The vertical dotted lines (black) indicate the recall rate of 0.5. The precision-recall pairs that correspond to $q$-value = 0.05 in each DE workflow are circled. $n = 1050$ cells were used for each test case.

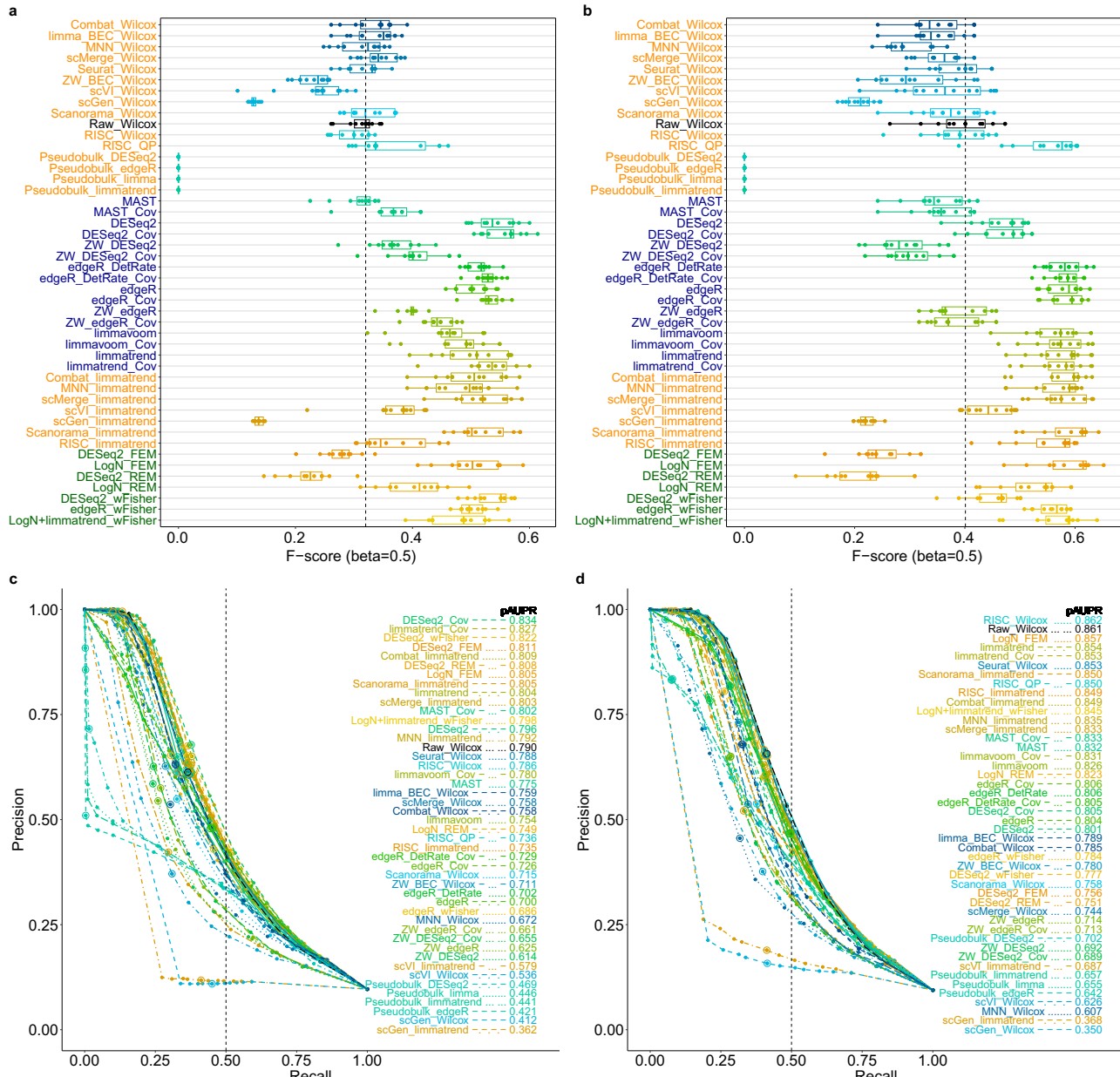

**Fig. 3 | Model-based simulation results for low depths (depth-10 and depth-4; zero rate >80%).** $F_{0.5}$-scores for 46 differential expression (DE) workflows for (**a**) depth-10 and (**b**) depth-4. Results for six cell proportion scenarios (12 instances in total: six for upregulated genes and six for downregulated genes) are represented as boxplots; the lower, center and upper bars represent the 25th, 50th and 75th percentiles, respectively, and the whiskers represent ±1.5 × interquartile range. The vertical dotted lines (black) indicate the median $F_{0.5}$-score of Wilcoxon test (Raw_Wilcox). Precision-recall curves for **c** depth-10 and **d** depth-4. The partial areas under the curve for recall rate <0.5 (pAUPRs) are computed and sorted in descending order in the legends. The vertical dotted lines (black) indicate the recall rate of 0.5. The precision-recall pairs that correspond to $q$-value = 0.05 in each DE workflow are circled. $n$ = 1000 cells were used for each test case.

were applied to edgeR and DESeq2 (denoted as ZW_edgeR and ZW_DESeq2, respectively), and specifically improved edgeR.

Next, we checked whether the integrative strategies truly improved the analysis of uncorrected data. First, the use of BEC data rarely improved DE analysis; one exception was scVI that considerably improved limmatrend. Second, covariate modeling (workflow names tagged with _Cov) overall improved the corresponding DE methods such as MAST, ZW_edgeR, DESeq2 and limmatrend for large batch effects. In particular, the performances of two single-cell-dedicated methods, MAST_Cov and ZW_edgeR_Cov were among the highest. However, covariate modeling tended to slightly deteriorate DE analysis for small batch effects. Third, meta-analysis methods did not improve on the naïve DE methods. Interestingly, DE analyses of pseudobulk

data, denoted as *pseudobulk methods*, showed good pAUPRs for small batch effects; however, they performed the worst for large batch effects. We also tested seven batches which yielded similar relative performances between DE workflows (Supplementary Fig. 2). With this increased number of batches, pseudobulk methods were rather improved, but their $F_{0.5}$-scores remained the lowest for large batch effects.

In recent years, shallow but high-throughput sequencing using for example 10x Genomics' technique has been widely used[34]. Therefore, we further performed simulation tests for low depths (average non-zero count of 10 and 4 after gene filtering, denoted as depth-10 and depth-4, respectively) (Fig. 3). As the depth was lowered, the use of observation weights of ZINB-WaVE deteriorated both edgeR and

DESeq2, because the low depth made it difficult to discriminate between biological zeros and technical zeros among the read counts[25]. The relative performances of Wilcoxon test and FEM for log-normalized data (LogN_FEM) were distinctly enhanced for low depths, whereas scVI improved limmatrend no more. For all depths, limmatrend, LogN_FEM, DESeq2, MAST and corresponding covariate models performed well and the use of BEC data rarely improved DE analysis. Covariate modeling overall improved DE analysis for large batch effects; however, its benefit was diminished for very low depths (depth-4).

## Model-free simulation tests

We devised a model-free simulation using real scRNA-seq data to incorporate realistic and complex batch effects and avoid potential bias toward parametric methods. First, we used the two batches from the human pancreatic data[35] (named as human1 and human2) that were produced by the same laboratory using the same sequencing platform (inDrop[36]). The alpha-cells were used for our simulation. Second, we used the two batches from Mouse Cell Atlas (MCA) that were produced by different laboratories using different sequencing platforms (Illumina HiSeq 2500[37] and NovaSeq 6000[38]). For MCA data, the T-cells were used for our simulation. Because these cell types contained several subtypes, the largest clusters that were matched between batches were selected for our simulation (see Methods). After removing sparsely expressed genes, the overall zero rates of the pancreatic alpha-cell and MCA T-cell data were 83 and 73%, respectively. Each batch dataset was randomly split into case and control groups with several different ratios, and then 20% of DE genes (10% up and 10% down) were simulated by downsampling positive counts in one group using binomial distribution (see Methods). The $F_{0.5}$-scores and precision-recall results for both data were shown in Supplementary Fig. 3. As expected, PVCA indicated small and large batch effects for the pancreatic and MCA data, respectively (Supplementary Fig. 3c, d). For the pancreatic data that had small batch effects and a low depth (Supplementary Table 1), most integrative strategies did not improve the DE analysis of uncorrected data, and limmatrend, DESeq2, edgeR as well as Wilcoxon test performed well with minor differences in pAUPR. The observation weights of ZINB-WaVE did not improve edgeR and DESeq2 for this low-depth data. However, for the MCA data that exhibited large batch effects and high depth in one batch, some integrative strategies and the use of observation weights were effective. For example, edgeR-based methods exhibited relatively low pAUPRs compared to other parametric methods; however, the weights of ZINB-WaVE considerably improved edgeR in pAUPR, and incorporating batch covariate further improved the method, rendering ZW_edgeR_Cov the top-performer in both $F_{0.5}$-scores and pAUPRs (Supplementary Fig. 3e–h).

Overall, both batch effects and sequencing depth had critical effects on scRNA-seq DE analysis in both model-based and model-free tests. For moderate depths, many parametric methods outperformed Wilcoxon test, and the observation weights and covariate modeling improved the parametric methods. Thus, ZW_edgeR_Cov, ZW_DESeq2_Cov and MAST_Cov were among the best performers. However, for very low depths, FEM, limmatrend and Wilcoxon test were the leading methods, and covariate modeling had limited effects even for large batch effects. For all depths, the use of BEC data rarely improved DE analysis for sparse data.

## Comparison of data distortions in DE analysis

From the simulation results, we counted the number of DE genes that reversed their signs by each DE workflow to compare the extent of data distortion. The signs of simulated DE genes declared by each DE workflow were compared with the known ground truth. For the *p*-value combination method (wFisher), the *p*-values for each batch were combined for both right- and left-tail directions and the sign for the

smaller combined *p*-value was used for each gene. Figure 4 and Supplementary Fig. 4 showed the proportions of DE genes that altered their signs by each DE workflow (referred to *error ratio*) for four different simulation results.

$$\text{error\_ratio} = \frac{\#\text{DE genes that altered their signs}}{\#\text{DE genes}} \times 100 \qquad (1)$$

Large error ratios indicated serious distortions for each DE workflow. Overall, limmavoom, pseudobulk_DESeq2, RISC_QP and the workflows that used BEC data tended to show relatively high error ratios, whereas Wilcoxon test and the parametric methods such as MAST, edgeR- and limmatrend-based methods yielded relatively accurate results. We then compared the error ratios among the significantly detected DE genes (*q*-value <0.05) (Fig. 4b, d and Supplementary Fig. 4b, d). Less than 50% of simulated DE genes satisfied this significance cutoff with which the numbers of incorrect sign prediction were dramatically reduced in most DE workflows. We additionally applied the FC threshold |log*FC*|>0.5 (base 2) to the significant DE genes. This threshold further reduced the number of detected DE genes substantially; however, the corresponding error ratios were only slightly reduced (Supplementary Fig. 5). Moreover, this FC threshold also reduced $F_{0.5}$-scores (Supplementary Fig. 6). These results indicated that using logFC threshold could help select a small number of reliable DE genes (or marker genes), but may not generally improve DE and function analysis of scRNA-seq data. For a low depth (depth-4), the error ratios were overall increased (Supplementary Fig. 4a, b). Distinctly high error ratios were observed for the workflows that used the deep-learning-originated BEC data (scGen and scVI), followed by those that used the observation weights of ZINB-WaVE.

Next, we specifically demonstrated the data distortions caused by the 10 BEC methods by comparing the logFC of DE genes before and after BEC without incorporating DE methods (Fig. 4e; Supplementary Fig. 4e). The logFC values were estimated using log-normalized count data. If a BEC method preserved the FC values, they would be aligned closely to the straight line $y = x$. Thus, we used the average angular (cosine) distance between each data point (DE gene) and this straight line to estimate the data distortion by each BEC method (see Methods). These angular distances for the six simulation scenarios were compared between 10 BEC methods (Fig. 4f, Supplementary Fig. 4f). For a moderate depth, Scanorama, ZW_BEC and MNN exhibited a relatively high-level distortion. For a low-depth (depth-4), the distortion level was overall increased, and scGen, ZW_BEC and scVI showed a high-level distortion. Notably, ZW_BEC enlarged FC levels, whereas Scanorama, MNN and RISC reduced them. scGen also enlarged FC levels for low depths. Overall, data distortions caused by BEC methods and DE workflows appeared worse for the lower-depth data and some BEC methods perturbed the FC values systematically.

## Effect of sparsity

We also compared the performance of DE workflows for less sparse data. We tested scRNA-seq data with approximately 40% zero rates and large batch effects for both moderate (depth-77) and low (depth-4) depths (Supplementary Fig. 7). Remarkably, many BEC methods and all the covariate models substantially improved the DE analysis. For both depths, ZW_edgeR_Cov and ZW_DESeq2_Cov were among the best performers. MAST_Cov and limmavoom distinctly performed well for moderate and low depths, respectively. The observation weight of ZINB-WaVE considerably improved edgeR for the moderate depth; however, it was less effective for the low depth. When small batch effects were tested, most BEC and covariate methods did not improve DE analysis, and pseudobulk_limma, pseudobulk_edgeR and Raw_Wilcox showed a good performance for both depths. Additionally, DESeq2 and LogN_FEM performed well for moderate depths, while

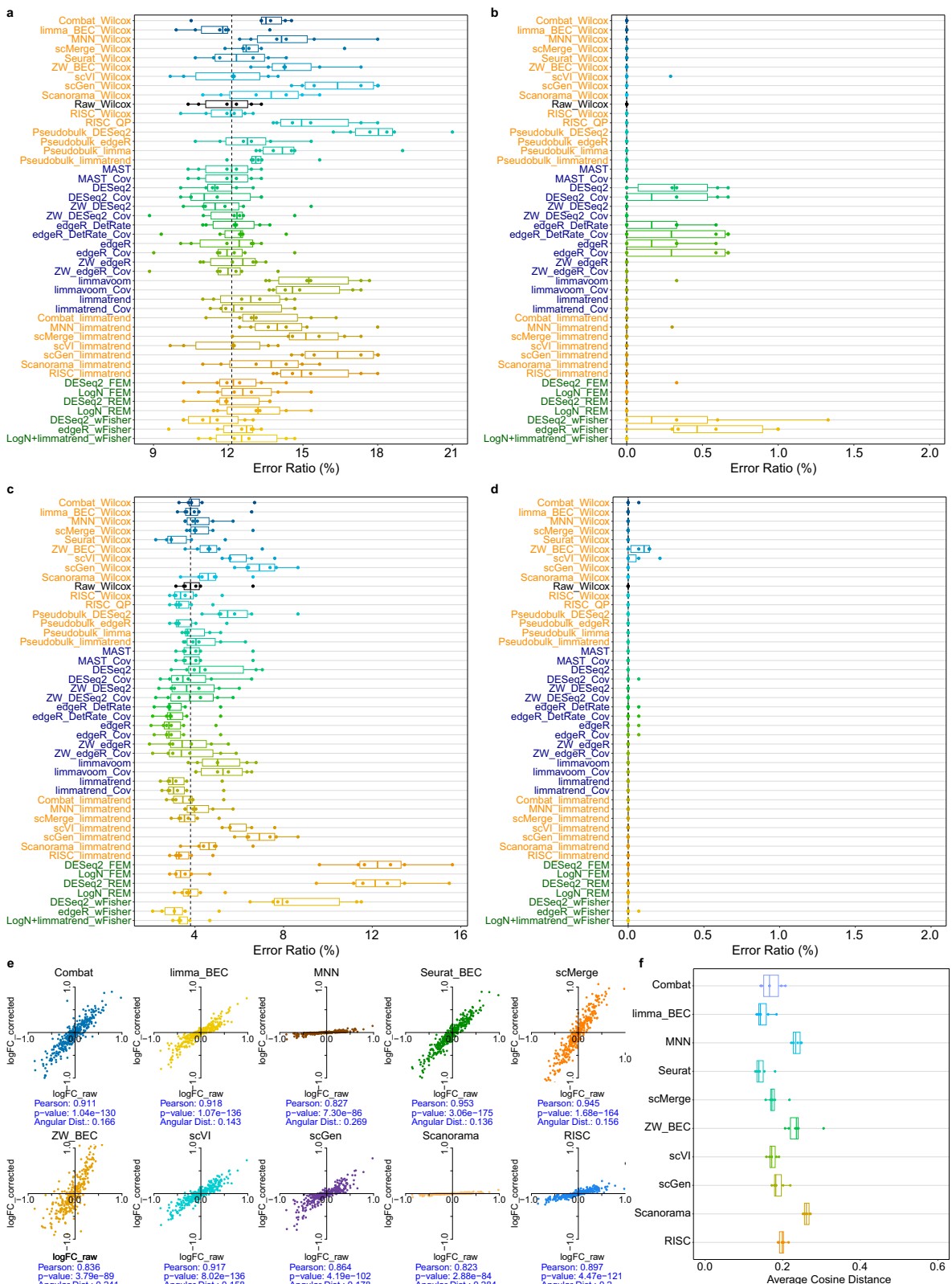

limmavoom/limmatrend and edgeR performed well for low depths (Supplementary Fig. 8).

## Control of false positives and false discoveries

Kim and colleagues[15] conducted a comprehensive analysis of scRNA-seq data for LUAD with over 200 K cells containing various cell types. We used the data for normal epithelial cells in the seven patients with

LUAD (stage I) to compare false-positives and false discoveries between DE workflows. The data for each patient were randomly split into two groups with several different ratios (2:8, 3:7, 4:6, and 5:5), and DE analysis was performed with no DE genes included. We repeated this experiment four times, and the numbers of genes with $p$-value <0.05 (false-positive) and $q$-value <0.05 (false discovery) were compared (Fig. 5a). edgeR_wFisher exhibited the worst false positive

**Fig. 4 | Distortion analysis for differential expression (DE) workflows.**
**a** Proportion of DE genes that altered their signs by each DE workflow (error ratios) for model-based simulation (two batches; large batch effects; depth-77). **b** Error ratios for the model-based simulation for only significantly detected DE genes (*q*-value <0.05). The vertical dotted lines (black) indicate the median error ratio of Wilcoxon test (Raw_Wilcox). **c** Error ratios for pancreatic alpha-cell (model-free) simulation data. **d** Error ratios for the pancreatic alpha-cell data for only significantly detected DE genes. **e** A scatterplot of the logFC values for the model-based simulation data with a moderate depth (depth-77) before (logFC_raw) and

after (logFC_corrected) applying batch-effect correction (BEC) methods: Combat, limma (limma_BEC), MNNCorrect, Seurat_BEC, scMerge, ZINB-WaVE (ZW_BEC), scVI, scGen, Scanorama and RISC. Pearson correlation, its *p*-value and the angular cosine distance (Angular Dist) of scatter plot are shown for each BEC method. **f** The distortion levels for the moderate depth data as measured by the angular cosine distance from the logFC scatterplot for six cell proportion scenarios. The lower, center and upper bars of each boxplot represent the 25th, 50th and 75th percentiles, respectively, and the whiskers represent ± 1.5 × interquartile range. *n* = 1050 cells were used in **a**, **b**, **e**, **f**, and *n* = 900 cells were used in **c** and **d**.

controls. edgeR and edgeR_DetRate also showed relatively poor controls of false positives and false discoveries, whereas ZW_edgeR improved the results. The poor false positive control of bulk RNA-seq tools in scRNA-seq DE analysis was also observed in the previous benchmark study for a single batch[28]. Two workflows that used BEC data (scGen_Wilcox and scGen_limmatrend) also exhibited poor false positive control. Other methods showed a reasonable control of false positives and false discoveries.

We then performed the same test for seven batches generated using model-based simulation (Fig. 5b) These data did not represent correlations between genes and had a higher depth (depth-77) compared to the LUAD scRNA-seq data. However, these two results exhibited some similarity: (1) poor controls of false positives and false discoveries using edgeR-related methods, especially edgeR_wFisher, (2) poor false positives controls of DE methods that used scGen BEC data, and (3) good controls of false-positives and false discoveries using Wilcoxon test, pseudobulk methods, MAST and ZW_edgeR. We note that Wilcoxon test yielded a number of false discoveries previously when "independent" samples where each batch contained either case or control condition only were analyzed[26]; however, it showed a reliable false discovery control when balanced samples were analyzed.

## Detection of known disease genes: lung adenocarcinoma

We used the cells from seven patients with LUAD (stage I)[15] to perform DE analysis between tumor and normal cells for three main cell types: epithelial cells, myeloid cells, and immune cluster composed of T lymphocytes and natural killer cells. These cell types together occupied 68.8% and 74.6% of normal and tumor cells in the LUAD scRNA-seq data, respectively (Supplementary Fig. 9a). Because true DE genes were not known for real data, we used the known lung cancer-related genes as the "standard positives". In total, 221 standard positive genes annotated with "adenocarcinoma of lung" were obtained from two disease gene databases, DisGeNET[39] and CTD[40]. These genes were weighted by the disease-association score (GDA score > 0.3) provided by DisGeNET (see Methods). All the genes analyzed were sorted by the DE *p*-values in each workflow, and the cumulative sum of GDA scores of standard positive genes, denoted as *cumulative score*, was compared between DE workflows in the respective cell types (Fig. 6a–c). In other words, we compared the weighted counts of known disease genes included in the top-*k* DE ranks to compare the performance of DE workflows.

To assess the ranks of known disease genes, we devised a truncated Kolmogorov–Smirnov (KS) test that only reflected the ranks of standard positives within the top 20% DE genes, with those in the remaining 80% forced to be evenly distributed. This approach can be particularly useful when selecting DE methods that are capable of prioritizing important genes in high ranks (see Methods), whereas the conventional KS test risks assessing a large number of middle ranks as significant[41]. Even with this conservative test, as many as 25 workflows exhibited significantly high ranks of the standard positives when epithelial cells were analyzed (*p*-value <0.01) (Fig. 6d). To further compare the performance of DE workflows, the area under the cumulative score curves for the top 20% DE genes, denoted as *pAUC*, was used. Many

workflows including RISC_QP, ZW_edgeR_Cov, edgeR_Cov, Raw_Wilcox and limmatrend_Cov exhibited similarly good pAUCs (Supplementary Fig. 10a). Covariate modeling and observation weights only marginally improved the corresponding parametric methods presumably due to the low depth of the data (average depth of 4.48 for epithelial cells). Interestingly, when myeloid cells and immune cluster were analyzed, none of the workflows showed significance (Fig. 6d).

We then performed DE analysis using the bulk RNA-seq data for LUAD from The Cancer Genome Atlas (TCGA)[42] comprising 493 cancer and 53 normal samples. The corresponding cumulative scores for the known disease genes were also represented in Fig. 6a–c. Remarkably, DE analysis of epithelial cells for only seven patients outperformed that of hundreds of bulk samples, demonstrating the high potential of scRNA-seq DE analysis to discover disease genes. Although the superiority of DE analysis of scRNA-seq data over that of bulk RNA-seq data has been expected, it has not been systematically analyzed. Here, we presented a statistical test comparing the performance of scRNA-seq and bulk sample DE analyses in detecting disease-related genes.

Figure 6e compared the ranks of 12 genes with high disease scores (GDA score > 0.5) for six selected DE workflows and four bulk sample analysis methods. The six workflows for scRNA-seq data detected the 12 genes with the average rank percentiles of 31.7% – 43.8% with ZW_DESeq2 performing the best, whereas much worse percentiles of 67.0% – 71.3% were obtained using the four TCGA analysis methods. In particular, *EGFR*, *KRAS*, *CTNNB1,* and *ERBB2* genes were captured within the top 20% rank by at least four scRNA-seq workflows, and the two genes *EGFR* and *KRAS*, which were most common in lung cancer, were ranked in the top 5.4% and 8.9% by ZW_edgeR_Cov, respectively. In contrast, none of the 12 genes were captured within the top 20% ranks in the analyses of TCGA data; specifically, *EGFR* and *KRAS* were only ranked 68.0% – 98.0% and 33.8% – 37.2%, respectively. These four genes were known to play important roles in the development of tumor malignancy related to RAS/RAF/MAPK and Wnt signaling pathways[43–45] (see Supplementary Notes). The top 20% DE genes for LUAD epithelial cells obtained using four selected DE workflows as well as TCGA analysis results were shown in Supplementary Data 1, which suggested novel LUAD-related genes.

We analyzed two more large-scale bulk sample expression datasets for LUAD that were obtained from GEO database[46] (GSE31210 and GSE43458), where analyses of scRNA-seq data still outperformed the analyses of these bulk sample data in detecting the known disease genes (see Methods and Supplementary Fig. 11). Furthermore, integrative DE analyses for all seven patients, except the pseudobulk methods, surpassed the analyses for individual patients (Supplementary Fig. 12).

## Detection of prognostic genes

Next, we performed the same analysis as above using another set of disease-related genes. These genes were selected from an integrated survival analysis of five microarray gene expression datasets for patients with LUAD (GSE29013, GSE30129, GSE31210, GSE37745 and GSE50081). The Cox proportional hazards model incorporating covariates of age, sex, and tumor stage[47] was applied to each dataset, and

the resulting $p$-values were combined for each gene using wFisher considering the signs of hazards ratios (HRs)[12]. These integrated $p$-values were adjusted for multiple testing correction, yielding 447 genes with $q$-value <0.05, denoted as "prognostic (standard positive) genes". We note that only seven of these genes were also included in the 221 known disease genes. Many DE workflows applied to epithelial cells except the pseudobulk methods detected the prognostic genes with significantly high ranks, and outperformed the analyses of TCGA data (Supplementary Fig. 13). Interestingly, several DE workflows applied to myeloid cells also detected the prognostic genes with significantly high ranks ($p$-value <0.01), suggesting the correlation of DE genes in those cell types with the survival of patients.

## Analysis of large-scale scRNA-seq data: COVID-19

We compared the performance of DE workflows for large-scale scRNA-seq data. Ren and colleagues have conducted a comprehensive analysis of scRNA-seq data from 196 patients with COVID-19[14]. We used the 48 patient samples that provided fresh/frozen PBMC samples. Among the cells from severe patients, the largest cell number was taken by monocytes (Supplementary Fig. S14a) which were known to play a crucial role in defending cells against viral infections[48]. We analyzed the 100,361 monocyte cells to benchmark DE workflows. We

performed DE analysis between mild/moderate and severe/critical symptoms assuming "sex" as two batch groups. "Age" has been regarded as an important factor for COVID-19 severity; however, 23 out of 24 senior patients (≥60 year old) had the severe/critical symptom which did not meet the balanced condition. We used the 133 genes annotated with the GO term, DEFENSE_RESPONSE_TO_VIRUS (GO:0051607) as standard positives (GO_Biological Process_2021)[49], and compared their ranks between DE workflows (Supplementary Fig. 14b). Among them, 27 workflows detected the standard positive genes with significantly high ranks by the truncated KS test ($p$-value <0.01). We tested pseudobulk methods by summarizing the counts for the 48 individual patients. However, the four pseudobulk methods did not exhibit significant results even using 48 samples. Whereas the analysis of pseudobulk data exhibited strict controls of false positives and false discoveries (Fig. 5), its predictive power for disease-related genes was not high in our analyses.

## Comparison of runtimes

The running times of the 46 DE workflows were compared in Fig. 7a for both LUAD and COVID-19 cases. These data represented moderate and large data sizes with 7764 and 100,361 cells with 7000–8000 genes. All the DE workflows were run on a Linux machine with AMD Ryzen

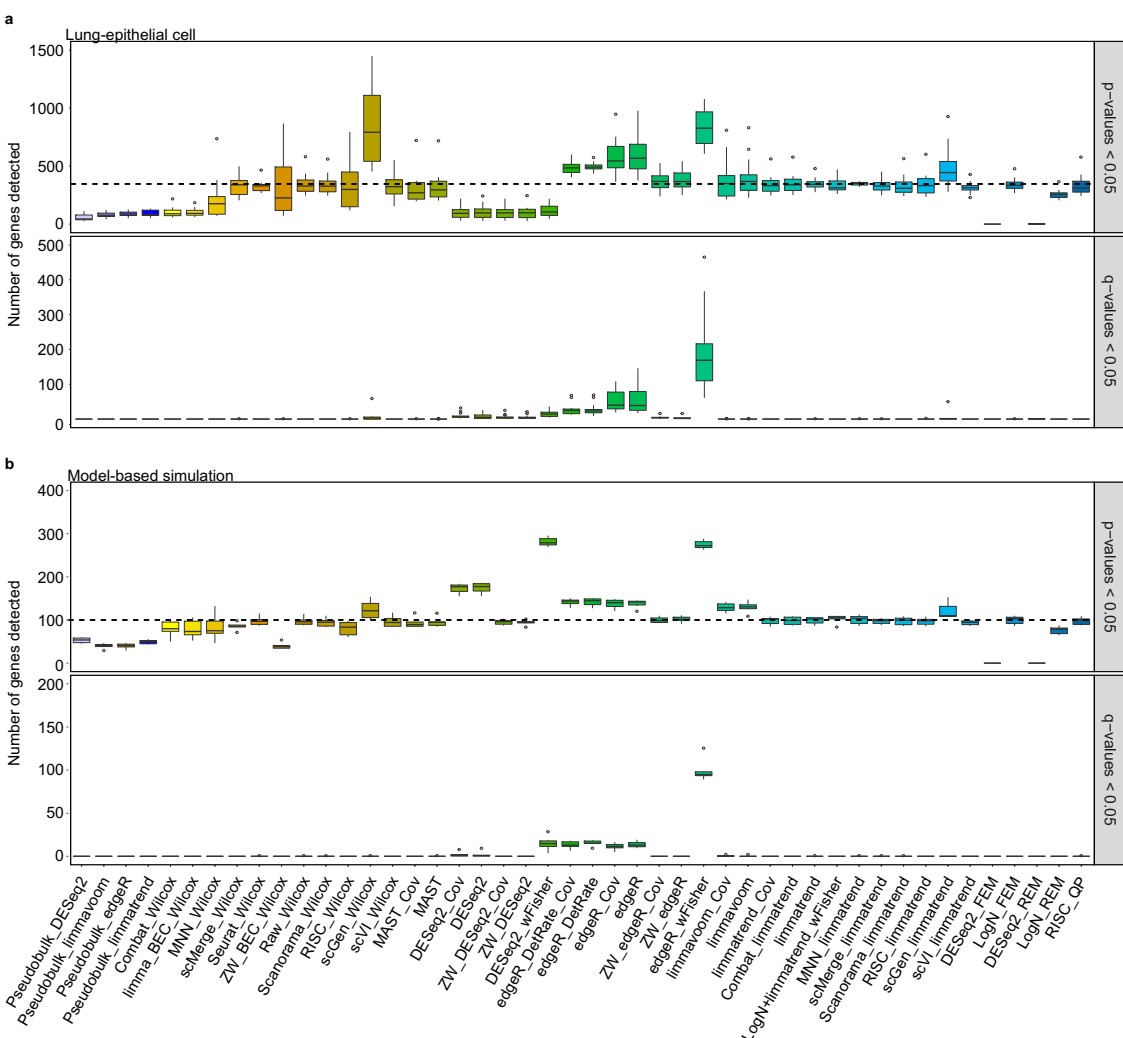

**Fig. 5 | Comparison of false positive controls ($p$-value <0.05) and false discovery controls ($q$-value <0.05) between differential expression workflows (gene filtering: zero rate > 0.95).** Test results for **a** seven batches of normal lung epithelial cells and **b** seven batches of model-based simulation data are shown as boxplots; the lower, center, and upper bars represent the 25th, 50th and 75th percentiles, respectively, and the whiskers represent ± 1.5 × interquartile range. Black dashes indicate the five percent of all genes tested. $n$ = 2331 and 2400 cells were used from normal lung epithelial cells and model-based simulation.

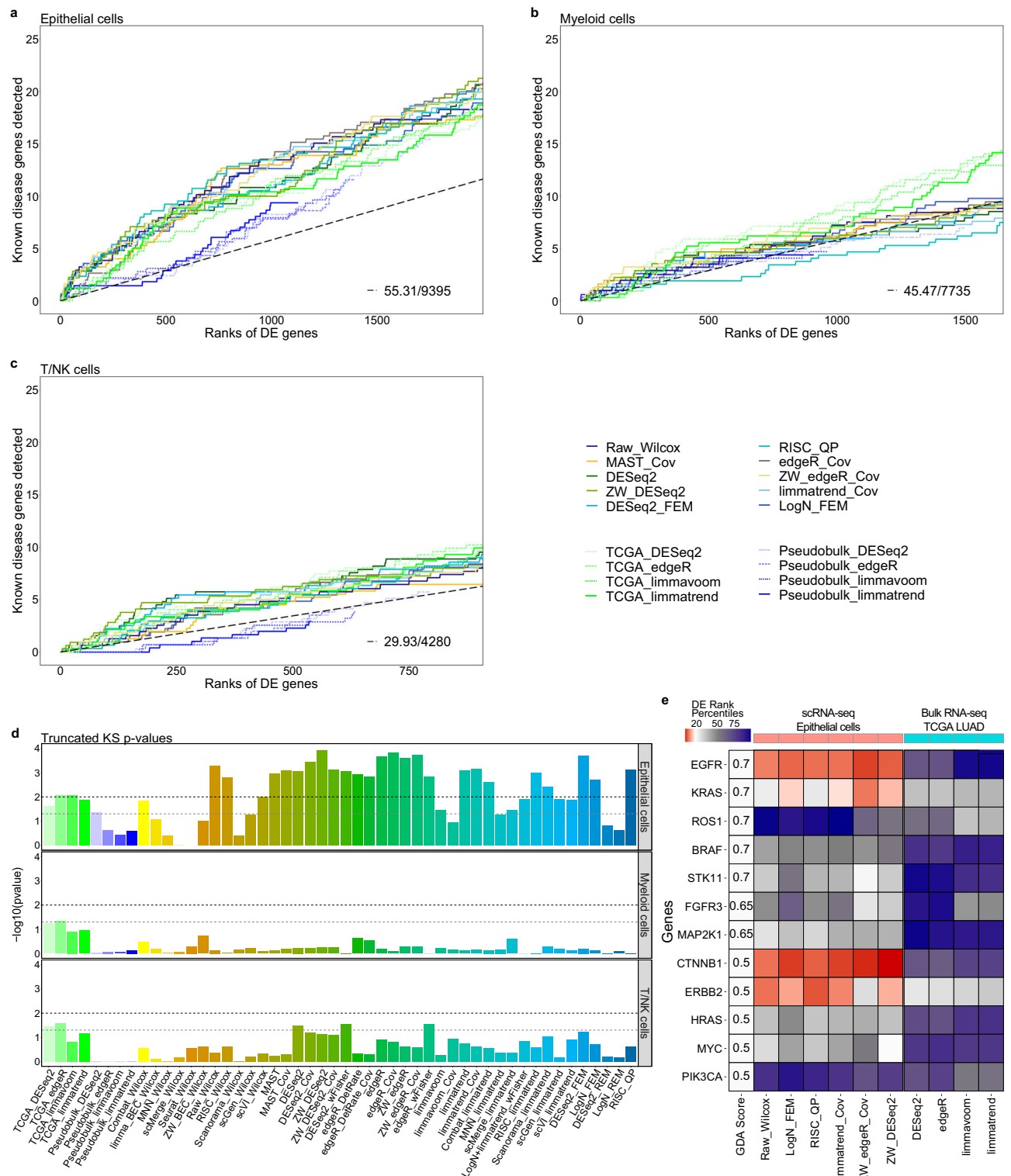

**Fig. 6 | Comparison of predictive powers for lung adenocarcinoma (LUAD) genes between differential expression (DE) workflows for scRNA-seq and bulk RNA-seq data.** Cumulative disease gene scores (GDA scores) for known disease genes up to top 20% DE gene ranks are shown for three cell types: **a** epithelial cells, **b** myeloid cells and **c** T/NK cells. X-axis represents the DE gene ranks in each DE analysis. Y-axis represents the cumulative score of known disease genes captured within top-$k$ gene ranks by each DE analysis. The black-dashed slopes represent the expected cumulative scores of known disease genes for random gene ranks. Ten and four methods are selected for analyzing scRNA-seq and bulk/pseudobulk data, respectively. **d** $p$-values of truncated Komogorov-Smirnov (KS) test for DE analyses of scRNA-seq and TCGA RNA-seq data are shown for the three cell types. Black and gray dashes represent the two significance cutoffs $p$-values = 0.01 and = 0.05, respectively. **e** Rank percentiles of the 12 known LUAD genes with GDA score no less than 0.5 are visualized for six DE workflows applied to LUAD epithelial cells and four bulk sample DE methods applied to TCGA LUAD data. $n$ = 7728, 17348, and 15293 cells were used for the analysis of epithelial, myeloid, and T/NK scRNA-seq data, respectively.

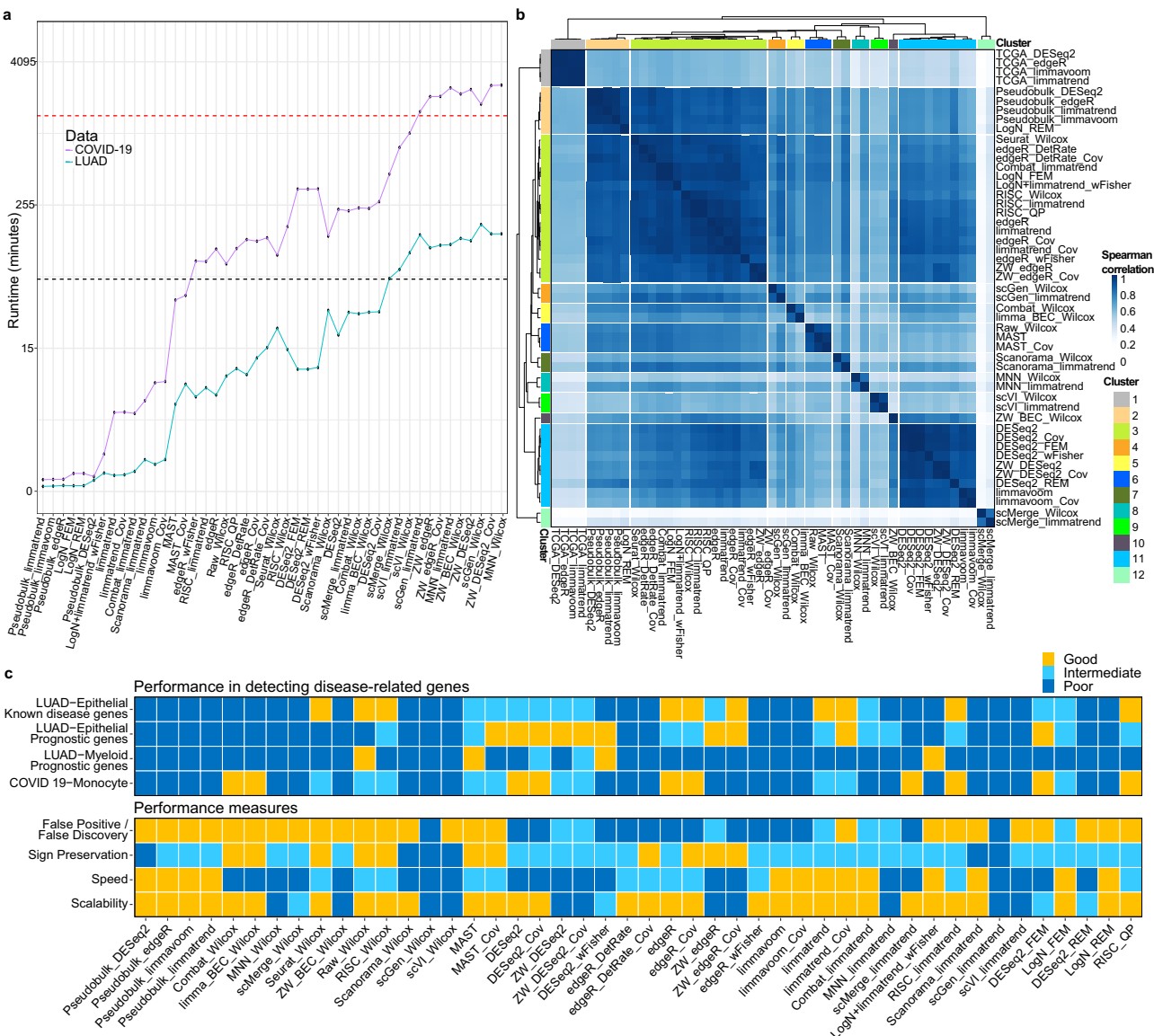

**Fig. 7 | Comparison of runtimes, similarity of differential expression (DE) workflows and other measures. a** CPU times (log-scale) of DE workflows taken for analyzing lung adenocarcinoma (LUAD) epithelial cell (*n* = 7728) and COVID-19 monocyte (*n* = 100361) scRNA-seq data, each containing 10278 and 7242 genes, respectively. **b** Clustering heatmap of 46 DE workflows for LUAD epithelial cell data and four DE methods for TCGA LUAD data. The Spearman rank correlation of DE genes were used as the similarity measure. **c** Comparison of the capability of prioritizing disease-related genes between 46 DE workflows and their performance classified in terms of false positive/discovery controls, sign preservation of DE genes, speed, and scalability.

Threadripper 3960 × 24-core processor and 128 Gb of DDR4 memory. The fastest were pseudobulk methods followed by limma-based workflows. Meta-analysis workflows exhibited shorter runtimes compared to naïve DE methods, as they applied a DE method to the individual batches. edgeR-based workflows showed intermediate runtimes. A long computation time was required for the workflows that used BEC such as MNN and scGen, and those that used the observation weight, which took longer than a day for COVID-19 data. Among the recommended, LogN_FEM and limmatrend took less than an hour even for COVID-19 data.

### Similarity between DE workflows
We compared the similarity of DE analysis results between the 46 workflows for LUAD epithelial scRNA-seq data and four DE methods applied to TCGA LUAD RNA-seq data. All 9,395 genes commonly found in both scRNA-seq and RNA-seq data were ranked by the signed log(*p*-value) score, -log(DE *p*-value) • sign(logFC) to compute Spearman rank

correlation between DE workflows. We used this score to sort genes, because log(FC) values for low-depth scRNA-seq data could be less reliable. The correlation heatmap and hierarchical clustering results were shown in Fig. 7b. The four TCGA analysis results formed a distinct cluster. The edgeR-, limmatrend- and RISC-based methods formed the largest cluster, followed by the cluster of DESeq2 and limmavoom methods. The four pseudobulk methods formed a separate cluster that was close to the largest cluster.

We also compared the similarity of DE workflows for COVID-19 monocyte data (Supplementary Fig. 15a). The DE workflows exhibited overall similar clustering patterns between LUAD and COVID-19 data. For example, pseudobulk methods formed a separate cluster; edgeR- and limmatrend-based methods were close to each other; and DESeq2- and limmavoom-based methods were also close. Besides, MAST and Raw_Wilcox were clustered together in both cases. The Baker's Gamma correlation indicated a high similarity between the two clustering results (0.56, *p*-value = 0)[50]. Even when we removed the covariate

workflows, the Gamma correlation remained high (0.42, *p*-value = 3.2e-04) (Supplementary Fig. 15c). We selected ten DE workflows from the three main clusters for LUAD data, and performed pathway analysis on the corresponding DE results in the next section.

## Pathway analysis for LUAD epithelial cell and TCGA data

We tested the pathway enrichments for scRNA-seq (epithelial cells) and TCGA data to compare the functional relevance of each DE analysis in cancer. Genes were ranked by their signed log(*p*-value) scores and the gene-set enrichment analysis (GSEA) was applied to the ranked genes in each DE workflow[51,52]. From the pathway database "wikipathway_2021"[49], 192 pathways that were most relevant to cancer progression were selected as standard positives. These pathways were selected on the basis of the ten oncogenic signaling pathways[53] and the seven cancer associated processes[54], as well as those including the keyword(s) tumor, cancer, or carcinoma in their names (see Methods). We classified these pathways into 16 categories for detailed interpretation of the GSEA results (Supplementary Data 2).

Interestingly, the analyses of scRNA-seq and TCGA data exhibited distinct functional categories. For example, "Ciliopathies (WP4803)" in the "cell polarity and migration" category ranked the first or second in all the ten scRNA-seq analyses, whereas it only ranked 21th to 62th in TCGA analyses. "Genes related to primary cilium development (based on CRISPR) (WP4536)", which belonged to the same category, was ranked the third in all the ten scRNA-seq analyses, whereas none of the TCGA analyses detected this pathway. These results represented the cell-type-specific perturbation of pathways in lung epithelial cells. Moreover, "VEGFA-VEGFR2 Signaling Pathway (WP3888)" in "angiogenesis" category was detected within top 10 ranks and the four categories "p53", "PI3K", "HIPPO" and "NOTCH" in oncogenic signalling pathways were also detected in scRNA-seq analyses. However, none of them were detected in TCGA analyses. In contrast, GSEA for TCGA data detected at least two and seven pathways in the two categories "genomic instability" and "inflammation", respectively, whereas GSEA for scRNA-seq data detected at most one and two in the respective categories. By focusing on the epithelial cell data, DE analyses of scRNA-seq data successfully detected many canonical oncogenic pathways as well as cell-type-specific pathways that the bulk sample analysis missed.

The scRNA-seq DE workflows selected from each cluster (Fig. 7b) exhibited distinct pathway analysis results. For example, Raw_Wilcox and MAST_Cov detected as many as six and seven pathways, respectively in "cell survival" category, but they detected none in "oxidative stress" category. Additionally, DESeq2, DESeq2_FEM, and ZW_DESeq2 also detected more pathways in "cell polarity and migration" and "oxidative stress" categories compared to other workflows, whereas they detected none in "p53" category. The GSEA results for four selected DE workflows and one TCGA analysis result are available from Supplementary Data 3.

## A gross performance comparison

The capability of prioritizing disease-related genes for the 46 DE workflows was categorized into three levels (Good, Intermediate and Poor) for both LUAD and COVID-19 scRNA-seq data (Fig. 7c). We used both criteria pAUC and truncated KS *p*-value to classify the workflows (see Methods). Pseudobulk methods, many DE methods that used BEC data, edgeR_DetRate, limmavoom and REM and *p*-value combining meta-analysis did not show a good performance. Raw_Wilcox, FEM meta-analysis and other parametric methods performed well, but the benefit of using observation weight was not clear for low depth-data. We also classified the DE workflows using other performance measures such as false positive/discovery controls, sign preservation of DE genes, speed and scalability (Fig. 7c). See Methods for detailed criteria in each measure.

## Discussion

Here, we benchmarked various workflows for DE analysis of scRNA-seq data with multiple batches, and investigated the impacts of batch effects, sequencing depth, and data sparsity on DE analysis. For sparse data (zero rate > 80%), the use of BEC data rarely improved DE analysis and the effect of using batch covariate depended on both batch effects and sequencing depth.

For a moderate depth (depth-77), single-cell-dedicated methods showed a good performance. For example, MAST which used zero-inflation model incorporating the sparsity information and edgeR combined with the observation weights for zero-inflation model (ZINB-WaVE) performed the best. Many parametric DE methods based on MAST, ZW_edgeR, DESeq2, and limmatrend surpassed the widely used Wilcoxon test, and covariate modeling of batch groups further improved the corresponding parametric methods for substantial batch effects. The bulk RNA-seq tool edgeR without observation weights exhibited relatively low precisions and poor false positive controls. For a low depth (depth-10), DESeq2, limmatrend and meta-analysis that used log-normalized data (LogN_FEM) showed a good performance, and covariate modeling still improved the results. However, the use of observation weights deteriorated edgeR and DESeq2 from this depth, as the low depth made it difficult for zero-inflation model to discriminate biological zeros from technical zeros. For an even lower depth (depth-4), covariate modeling hardly improved DE analysis and observation weights had deleterious effects on DE analysis. In this case, LogN_FEM and two naïve methods, limmatrend, and Wilcoxon test were among the best performers. Indeed, in our analysis of LUAD and COVID-19 scRNA-seq data that had a low depth (4.48 and 3.27, respectively), covariate modeling only marginally improved the corresponding parametric methods in detecting disease-related genes. While the observation weight only marginally improved edgeR and DESeq2 in LUAD data analysis, it deteriorated the methods in COVID-19 data analysis. We also tested pseudobulk methods which exhibited good false discovery controls and overall discriminatory abilities in a recent study[26]. We observed similar good results with pseudobulk methods for small batch effects; however, they were highly vulnerable to batch effects and exhibited low sensitivities in our tests.

Moreover, we compared the signs of DE genes declared by each DE workflow with the ground truth to estimate the data distortions caused by each workflow. For a moderate depth, many workflows that used BEC, RISC_QP, and limmavoom exhibited relatively high error ratios. For a low depth, the overall error ratios were increased, and relatively high error ratios were observed with the workflows that used the observation weights and the BEC data obtained from deep-learning methods (scGen and scVI). We further examined how BEC methods affected scRNA-seq data before applying a DE method. The analysis results demonstrated that BEC methods introduced additional perturbations (or noise) to data as well as systematic changes in the FC values. Whether these perturbations were beneficial or not could only be tested by comparing the performance of DE workflows in identifying DE genes and their signs. Our tests showed that the artifacts introduced by BEC methods outweighed their benefits in DE analysis, especially for sparse and low-depth data.

For less sparse data (zero rate ≈40%), the situation changed much; many BEC methods considerably improved DE analysis and covariate modeling clearly improved DE analysis for both moderate and low depths. Moreover, many workflows including Wilcoxon test showed good overall discriminatory abilities (pAUPR) with minor differences. This showed the performance of DE workflows depended on batch effects, sequencing depth, data sparsity, as well as their interactions, posing a challenge to choosing an optimal DE workflow for various conditions. Thus, we suggested suitable DE workflows under different conditions based on our simulation and real data analysis results (Table 1). Here, the sequencing depth and sparsity were explicitly given

**Table 1 | Recommended differential expression (DE) workflows for different experimental conditions**

| Sparsity (zero rate) | Depth* | Batch Effects | Recommended DE workflows |
|---|---|---|---|
| 80% | 77 | Substantial | MAST_Cov, ZW_edgeR_Cov, ZW_DESeq2_Cov, scVI_limmatrend, DESeq2_FEM, limmatrend_Cov |
| 80% | 77 | Small | MAST, ZW_edgeR, ZW_DESeq2, Pseudobulk_limma, DESeq2_FEM, limmatrend_Cov |
| 80% | 10 | Substantial | DESeq2_Cov, limmatrend_Cov, DESeq2_wFisher, LogN_FEM, MAST_Cov, Raw_Wilcox |
| 80% | 10 | Small | DESeq2, limmatrend_Cov, LogN_FEM, Pseudobulk_edgeR, Pseudobulk_limma, Raw_Wilcox |
| 80% | 4 | Substantial | LogN_FEM, limmatrend, Raw_Wilcox, RISC_QP |
| 80% | 4 | Small | LogN_FEM, limmatrend, Raw_Wilcox, RISC_QP |
| 40% | 77 | Substantial | MAST_Cov, ZW_edgeR_Cov, ZW_DESeq2_Cov, limma_BEC_Wilcox, Scanorama_limmatrend, logN_FEM |
| 40% | 77 | Small | Pseudobulk_limma, Raw_Wilcox, Pseudobulk_edgeR, DESeq2, LogN_FEM |
| 40% | 4 | Substantial | limmavoom_Cov, limmatrend_Cov, ZW_edgeR_Cov, ZW_DESeq2_Cov, logN_FEM, limma_BEC_Wilcox |
| 40% | 4 | Small | Pseudobulk_limma, limmvoom, limmatrend, Pseudobulk_edgeR, Raw_Wilcox, edgeR |

In 40% sparsity cases, recommended methods were selected based on simulation results only.
*Average nonzero count in each cell after filtering sparsely expressed genes (zero rate > 0.95).

from the data in hand, whereas batch effects should be estimated from the study design, data distribution or using PVCA. For example, if the batch data were obtained from different research groups or sequencing protocols, we might expect sizable batch effects. This could be affirmed by using commonly used dimension reduction and visualization techniques[3] and assessed using batch effects quantification tools[55]. PVCA required quite a long computation time for large-scale data; however, the time was greatly saved by using a randomly selected subset from scRNA-seq data.

Finally, we tested whether DE workflows for multi-batch scRNA-seq data could be used to prioritize disease-related genes better than analyses of bulk or pseudobulk data. This was verified for two independently derived sets of cancer-related genes and three large-scale bulk sample datasets for LUAD. For a specific cell type (epithelial cells), many DE workflows applied to scRNA-seq data exhibited a superior predictive power for the cancer-related genes compared to the analyses of bulk or pseudobulk data. Furthermore, we tested DE workflows for large-scale COVID-19 scRNA-seq data. Each individual in these data belonged to multiple biological or technical categories, such as age, sex, sequencing protocol, sample processing method and cohort region. These factors imposed large and complex group effects on the data, which could have deteriorated pseudobulk methods. Many of those factors did not meet the balanced condition when the whole data were considered for DE analysis; thus, we took the most universal factor as a batch category to test DE workflows for large-scale data. Many DE workflows successfully detected virus-related genes in their significantly high ranks. Our results suggested using integrative DE analysis of scRNA-seq data considering cells as independent replicates rather than using bulk or pseudobulk data to discover disease-related genes.

## Methods

### $F_\beta$-score and partial area under precision-recall curve

In DE analysis of scRNA-seq data, it is often important to identify a small number of genes (markers) that are capable of characterizing each cell type. Moreover, it is not reasonable to expect to identify all DE genes from highly noisy and sparse data. Thus, we use generalized F-score ($F_\beta$) and partial AUPR (pAUPR) that weigh precision twice higher than recall to assess a DE analysis method. In binary classification task, F-score is the harmonic mean of precision and recall. For a list

of DE genes (q-value <0.05), we use $F_\beta$ ($\beta = 0.5$) defined as follows:

$$F_\beta = \frac{(1+\beta^2) \cdot \text{precision} \cdot \text{recall}}{\beta^2 \cdot \text{precision} + \text{recall}}, \quad \beta > 0 \tag{2}$$

The $F_\beta$-scores were calculated for both up and downregulated genes and both results were used. To assess the general performance of a classifier, precision-recall curve has often been considered. Instead of using the whole AUPR, we suggested using pAUPR ($T = 0.5$) defined as follows:

$$pAUPR_T = \frac{1}{T} \int_0^T \text{precision}_t \, dt, 0 < T < 1 \tag{3}$$

### Acquisition and preprocessing of gene expression data

The LUAD TCGA data were downloaded from UCSC xenabrowser (https://xenabrowser.net/datapages/). HT-seq count data and gene/mapping data were used for DE analysis. The curated gene sets were downloaded from Enrichr Gene-set Library (https://maayanlab.cloud/Enrichr/#libraries). "WikiPathway_2021_Human" was used for pathway analysis and the 133 genes annotated with "defense response to virus" were obtained from "GO_Biological_Process_2021". LUAD and COVID-19 scRNA-seq data (GSE131907 and GSE158055, respectively) and the microarray data for LUAD bulk samples were downloaded from NCBI Gene Expression Omnibus database with their accession numbers (GSE43458, GSE29013, GSE30129, GSE31210, GSE37745, GSE50081).

For both simulation and real scRNA-seq data, genes expressed in less than 5% of the cells analyzed were excluded. For both LUAD and COVID-19 scRNA-seq data, the cells with high mitochondrial gene expression were removed using the PercentageFeatureSet function from the Seurat R package; the same thresholds used in the original studies 20 and 10% were applied to the three cell types of LUAD and monocyte of COVID-19 data, respectively. Then, we disabled the additional filtering in each DE analysis method. Specifically, we set the logFC threshold and minimum number of expressed cells to zero. Several BEC methods, such as Seurat and scMerge have used highly variable genes only which yielded corrected data with a low dimension. Thus, we used all the genes for

those methods to have the corrected data with the original high dimension to test downstream DE analysis. For bulk sample data, we used biomaRt R package to map probe IDs to GRCh38 gene symbols and used the protein coding genes. The ENSEMBL IDs of TCGA data were converted to GRCh38 gene symbols.

For each DE workflow, either raw count or log-normalized count data were used as input data as recommended for each DE method used. All the 10 BEC methods yielded batch-corrected data in their log-scale, which were directly used for Wilcoxon test and limmatrend. DESeq2- and edgeR-based workflows used raw count data as input. ZINB-WaVE yielded both the observation weights and corrected data, and the former was used as additional input for edgeR and DESeq2 (ZW_edgeR and ZW_DESeq2). Detailed preprocessing (e.g., log-normalization) of the input data for each DE workflow was described in Supplementary Notes.

## Model-based simulation for 80% and 40% zero rates

Splatter R package[32] was used to simulate scRNA-seq data based on negative binomial model. The dropout parameter values *dropout.mid* = 0.01– .05 and 3.7–3.8 were used to simulate data with overall zero rates 40% and 80%, respectively. *splatSimulate* function was used to simulate different batches. Large batch effects (*batch.facLoc* = 0.4 and *batch.facScale* = 0.4) and small group differences (*de.facLoc* = 0.2 and *de.facScale* = 0.2) were used to simulate large batch effects. We created six scenarios for combinations of two dropout values and three group size ratios (2:8, 3:7, 4:6). The batch sizes with 300 and 750 cells were used for the two-batch case; and 100, 150, 200, 300, 400, 500, and 750 were used for the seven-batch case. Approximately 2500 genes survived gene filtering and included 20% DE genes (half up and half downregulated). No statistical method was used to predetermine sample size.

## Model-free simulation

MCA and pancreas data were used to simulate scRNA-seq data. MCA data comprised two independent datasets obtained using different sequencing techniques. The original data included two batches containing 4,239 and 2,715 cells. We chose T-cells for our simulation. Because T-cells included several subtypes, we selected the largest clusters from each batch that shared marker genes identified by using FindMarkers function in Seurat package[19]. Specifically, we selected clusters with 358 and 266 T-cells from different batches. For pancreas data, we used the clusters with 241 and 659 alpha-cells from human1 and human2 batches, respectively. Then, each batch dataset was randomly divided into case and control groups with different ratios to cover several scenarios. We then randomly selected two groups of genes, each with 10% of all genes; one group was downsampled in the case group and the other downsampled in the control group using binomial distribution to simulate DE genes. The success probability for the binomial distribution was sampled from the beta distribution with the shape parameters $\alpha = \beta = 2$ that are expected to generate DE genes with the median fold-change two.

## Covariate modeling

The log-linear model[56,57] has been frequently used to model the read count data as follows:

$$\log\left(E\left(y_{ij}\right)\right) = \alpha_{i0} + \alpha_{i1}L_j + \sum_{b=1}^{B} \beta_{jb}I_{jb} + \sum_{g=1}^{G} \gamma_{jg}I_{jg} \quad (4)$$

where $y_{ij}$ is the read count of gene $i$ and sample $j$, $L_j$ is the library size of sample $j$, $I$ is the indicator function of a specific sample group, $\alpha$'s, $\beta$'s, and $\gamma$'s are the model parameters, $B$ and $G$ are the numbers of batches and sample groups used, respectively. Then, DE of a gene can be tested using quasi-likelihood ratio, Wald or moderated $t$-test under a logarithmic count model incorporating the batch variable[9,29,30,56,58].

## Principal variance component analysis

Principal variance component analysis (PVCA)[33] was used to estimate the variability of experimental effects. PVCA combines principal component analysis (PCA) and variance components analysis (VCA) to take advantages from both techniques. PCA reduces the dimension of data while preserving their major variability. VCA fits a mixed linear model using the factors of interest to estimate and partition the total variability. Whereas PVCA is a generic approach used to quantify the proportion of variations of different effects, it provides a handy assessment for the batch effects before and after the correction.

## Estimation of logFC values and data distortion from scRNA-seq data

For a read count matrix $[c_{ij}]$ for gene $i$ and cell $j$, the log-normalization of $c_{ij}$ is defined as follows:

$$\text{lognorm}\left(c_{ij}\right) = \log\left(\frac{c_{ij}}{L_j} * 10^4 + 1\right) \quad (5)$$

where $L_j$ is the library size (total count) of cell $j$. Then, the log2 FC value for gene $i$ between case and control conditions was estimated as follows:

$$\log FC_i = \frac{1}{n'_{c'_{ij} \neq 0}} \sum_{c'_{ij} \neq 0} \text{lognorm}(c'_{ij}) - \frac{1}{n_{g_{ij} \neq 0}} \sum_{c_{ij} \neq 0} \text{lognorm}(c_{ij}) \quad (6)$$

where $c'_{ij}$ and $c_{ij}$ are read counts of gene $i$ for the case and control groups, respectively. We compared the logFC values before and after BEC. We used the average angular (cosine) distance between each data point (DE gene) and the straight line y = x (Fig. 4e) as the measure of data distortion by each BEC method.

$$\text{Data distortion(BEC)} = \frac{1}{n} \sum_{i=1}^{n} \left[ 1 - sign(\log FC_i^{raw}) \cdot \frac{\left\langle (1,1), \left( \log FC_i^{raw}, \log FC_i^{corrected} \right) \right\rangle}{\sqrt{2} \cdot \| \left( \log FC_i^{raw}, \log FC_i^{corrected} \right) \|_{L_2}} \right] \quad (7)$$

## Collection of known disease genes

Two disease gene databases, DisGeNET and CTD were used to retrieve known lung cancer genes. In DisGeNET, 2438 genes were annotated with term, "Adenocarcinoma of the lung (disorder)". DisGeNET provides gene-disease association score (GDA score), which is weighted sum of the number of each level/type of sources, and the number of publications supporting the association. Among the 2438 genes, we have selected only 207 genes with GDA score 0.3 or larger. In CTD, we have selected 158 genes that were annotated with "Adenocarcinoma of Lung" and curated as "Marker/mechanism" in "Direct.Evidence" field. Among them, 144 genes were also selected from DisGeNET and their median score was given to the rest 14 genes that were exclusively selected from CTD. In total, 221 genes were used as standard positives.

## Categorization of standard positive pathways in lung cancer

The standard positive pathways were categorized on the basis of ten oncogenic signaling pathways[53] and seven cancer-associated processes[54]. The ten oncogenic signaling pathways included (1) cell cycle, (2) Hippo signaling, (3) MYC signaling, (4) NOTCH signaling, (5) oxidative stress response/NRF2, (6) PI-3-Kinase signaling, (7) receptor-tyrosine kinase (RTK)/RAS/MAP-Kinase signaling, (8) TGFβ signaling, (9) P53 and (10) β-catenin/WNT signaling. Here, "MYC signaling" category was not detected by any analysis, so was excluded. The seven cancer-associated processes included (1) cell proliferation, (2) cell polarity and migration, (3) cell survival, (4) cell metabolism, (5) cell fate and differentiation, (6) genomic instability, and (7) tumor microenvironment. Among them, "cell cycle", "cell proliferation" and "cell

fate and differentiation" were combined into one category, and "tumor microenvironment" was divided into its subcategories, "inflammation" and "angiogenesis". Lastly, pathways that included the keywords, tumor/cancer/carcinoma in their names were collected into a separate category, where less relevant pathways such as retinoblastoma or glioblastoma were excluded. In total, 190 standard positive pathways for cancer were classified into 16 categories (Supplementary Data 2).

### Analysis of LUAD bulk-sample expression data

DE analysis for TCGA LUAD bulk RNA-sequencing data between 493 cancer and 53 normal samples were performed incorporating covariates age, sex and smoking history using four methods, DESeq2, edgeR, limm, and limmatrend methods. 16,129 genes with five or larger mean count that were commonly found in gene-filtered epithelial scRNA-seq data were analyzed. Two LUAD microarray expression datasets (GSE31210 and GSE43458) were also analyzed. The former consisted of 226 tumor and 15 normal samples and covariates of age, sex, and smoking history were incorporated in DE analysis. The data were normalized by MAS5 and the log-normalized data were used for limmatrend. The latter consisted of 80 cancer and 30 normal samples. Only smoking history was available and used as covariate. RMA normalization and limmatrend were used.

### Criteria for classifying performance

**Standard positive gene detection.** We used the ranks of pAUC and truncated KS $p$-values to classify the performance into three categories as follows:

- Good: Truncated KS $p$-value < 0.01 and top 10 in pAUC
- Intermediate: Cases other than Good and Poor
- Poor: Truncated KS $p$-value > 0.01 or pAUC rank > 20

**False Positive/False Discovery.** We used the number of false positives and false discoveries to classifiy the performance into three categories as follows

- Good: Zero median false discovery and the median number of false positives not larger than 5% of analyzed genes for both low- and moderate-depth data (Fig. 5).
- Intermediate: Cases other than Good and Poor
- Poor: Median false discoveries larger than zero for both low- and moderate-depth data AND median number of false positives larger than 5% for either low- or moderate-depth data.

**Sign preservation.** Because the ranges of values were different between the results of datasets, we think of aggerating the relative difference between boxplots/groups of performance values of methods. The percentage of errors (P) is calculated based on the difference between medians (DBM) and the overall visible spread (OVS) as:

$$P = \frac{DBM}{OVS} \times 100 \qquad (8)$$

- Good: $P < 30\%$
- Intermediate: between Good and Poor
- Poor: $P > 60\%$

**Speed.** We used LUAD epithelial cell and COVID-19 monocyte data to compare the computing times between DE workflows and classified them based on their ranks as follows:

- Good: Runtime of COVID-19 < 10 mins
- Intermediate: between Good and Poor
- Poor: Runtime for LUAD data > 30 mins or runtime for COVID-19 > three hours

**Scalability.** We compared the proportionality between the computing time and the data size. We estimated this coefficient for the square root of the number of data entries (cells × genes). For dataset i including $N_i$ cells and $M_i$ genes, the computing time $T_i$ (seconds) of method K was modeled as

$$T_i = \alpha_K \sqrt{N_i \cdot M_i} \qquad (9)$$

The scalability of method K was classified based on the coefficient $\alpha_K$ as follows:

- Good: $\alpha_K < 1$
- Intermediate: between Good and Poor
- Poor: $\alpha_K > 2$

### Truncated Kolmogorov-Smirnov test

Kolmogorov-Smirnov (KS) test assesses the maximum distance between empirical and null cumulative distribution functions (cdf). The empirical distribution was generated by accumulating the gene scores of standard positives in the order of DE gene $p$-values and the test statistic is given as follows:

$$F_x(u) = \frac{\sum_{i=1}^{u} w_i}{\sum w_i} = \text{cdf of empirical distribution} \qquad (10)$$

$$F_y(u) = \text{cdf of null hypothesis} \qquad (11)$$

**KS statistic (right-tailed).** $D^+ = max(F_x(u) - F_y(u))$ where $w_i$'s are the weights of standard positive genes. If the *ith* gene does not belong to standard positive genes, $w_i = 0$.

A drawback of KS test is that the maximum discrepancy $D^+$ can occur for a low gene rank[41]. Because we are interested in methods that are capable of prioritizing standard positive genes in high ranks, we modified the statistic so that $D^+$ can occur only within top 20% ranks as follows:

$$\text{wKS statistic(right − tailed)}: \widetilde{D}^+ = max(\widetilde{F_x}(u) - F_y(u)) \qquad (12)$$

$$\widetilde{F_x}(u) = \begin{cases} F_x(u), & u < N \\ F_x(N) + \frac{(u-N)}{(u_{max}-N)} \cdot (1 - F_x(N)), & u \geq N \end{cases} \qquad (13)$$

where $u_{max}$ is total number of genes $N$ corresponds to the top 20% rank. In other words, the ranks of standard positives outisde the top 20% DE genes were uniformized not to affect the test result.

### Reporting summary

Further information on research design is available in the Nature Portfolio Reporting Summary linked to this article.

## Data availability

The single-cell, bulk sample and pathway data used in this study are available publicly and described in the **Methods** section. The scRNA-seq and microarray data were downloaded from the GEO database[46] through their accession numbers (scRNA-seq LUAD: "GSE131907"; scRNA-seq COVID-19: " GSE158055"; microarray LUAD: "GSE29013", "GSE30129", "GSE31210", "GSE37745", "GSE43458" and "GSE50081"). TCGA LUAD RNA-seq data were downloaded from the UCSC xenabrowser (https://xenabrowser.net/datapages/?dataset=TCGA-LUAD. htseq_counts.tsv&host=https%3A%2F%2Fgdc.xenahubs. net&removeHub=https%3A%2F%2Fxena.treehouse.gi.ucsc.edu% 3A443). Pathway data (WikiPathway_2021_Human and GO_Biological_-Process_2021) were downloaded from Enrichr Gene-set Library[49] (WikiPathway_2021_Human: https://maayanlab.cloud/Enrichr/ geneSetLibrary?mode=text&libraryName=WikiPathway_2021_Human; GO_Biological_Process_2021: https://maayanlab.cloud/Enrichr/

geneSetLibrary?mode=text&libraryName=GO_Biological_Process_2021). The collection of known disease genes were downloaded from two public databases (DisGeNET: https://www.disgenet.org/; CTD: http://ctdbase.org/). All other relevant data supporting the key findings of this study are available within the article and its Supplementary Information files or from the corresponding author upon reasonable request. Source data are available at Zenodo (https://doi.org/10.5281/zenodo.7645614[59]).

## Code availability

The R and Python codes used for our simulation tests are available at both GitHub (https://github.com/noobCoding/Benchmarking-integration-of-scRNAseq-differential-analysis) and Zenodo (https://doi.org/10.5281/zenodo.7608396).

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

## Acknowledgements
We would like to thank Dr. Hae-Ock Lee for providing information and advice on analysis of the LUAD scRNA-seq data. This work was supported by the National Research Foundation (NRF) of Korea: Basic Science Research Program [NRF-2020R1A2C2102268 (D.N.)] and Genomics Programs [2020M3C9A5086069 (D.N.); 2016M3C9A3945893 (D.N.)].

## Author contributions
H.C.T.N., B.B., and S.Y. performed simulation tests and analyzed real data. D.N., H.C.T.N., and B.B. devised the study and wrote the manuscript. T.P. reviewed and edited the manuscript. D.N. supervised the project.

## Competing interests
The authors declare no competing interests.
