## [Peer Review File · Nature Communications]

REVIEWER COMMENTS

Reviewer #1 (Remarks to the Author):

This manuscript benchmarked different strategies of post-integration differentially gene expression (DEG) analysis with simulated and real single-cell data and recommended parametric DEG methods such as edgeR with batch information as covariate. The manuscript also mentioned that batch effect correction introduces new errors and data distortion and thus DEG analysis on batch corrected gene expression is not recommended.

Comments:

1. It would be helpful to demonstrate the distortion introduced to the high-dimensional gene expression data by the batch correction methods, and whether all batch correction methods introduce such distortion. This will help explain why batch corrected values deteriorate the DEG results.
2. Simulated DEGs and known disease or prognostic genes were used as ground truth and F1 score and AUC curves were used to quantify the accuracy of DEGs. I wonder whether the magnitude of differential expression (i.e. log fold change between case and control) is taken into consideration. Do DEG methods influence the genes with large and subtle differences equally or they effect genes with subtle differences more? Likewise, do DEG methods effect highly and lowly/rarely expressed genes equally? A gene that is lowly or rarely expressed should be excluded for DEG analysis even if it has significant p values and large log fold change.
3. Naming of some of the DEG strategies is a bit difficult to follow, for example, ZW could mean the observation weights or corrected gene expression values from Zinb-Wave. I was not sure if raw or corrected values were used for DEG for certain methods.
4. Have the authors tried DEG identification within each batch and compare the results with those from integrative DEGs, then count the frequency of integrative DEGs appearing in individual batch DEG sets. The higher frequency suggests higher confident level for that gene. This is useful when large batch effects are present; no ground-truth DEGs are available; in particular, when the difference between the groups (case vs control) is subtle.
5. In the section of Preservation of signs of differential expression, I feel that the effects of BEC and DEG are entangled. It would be simpler to first look at log fold change without any DEG methods, and compare log fold change before and after BEC.

Reviewer #2 (Remarks to the Author):

Summary

The authors present a benchmarking study on differential expression (DE) analysis methods in single-cell RNAseq and address whether batch-corrected single-cell data can or should be used for DE analysis. The question is timely and addresses the need for accurate DE analysis. While it has been theoretically addressed that DE analysis should not be done on integrated data as they may contain technical artifacts introduced by the batch effect correction (BEC) method, the actual impact has not been systematically addressed. The authors then apply DE analysis to several publicly available single-cell RNAseq datasets

and employ a comparison to bulk methods (TCGA and microarray data). Here, the statement that bulk RNA-seq is outperformed by single-cell RNA-seq is not new. Overall, DE analysis on BEC data is not more accurate than DE analysis on uncorrected data. Importantly, including the batch covariate in the DE analysis generally improved the accuracy of the results. However, any theoretical explanation on these results is missing. Also, the authors could substantially contribute to show whether pseudobulk methods may be useful for DE analysis, but the synthetic benchmark does not include pseudobulk methods. I think that the approach of the manuscript is valid, but the work has several shortcomings that limit the power of the results.

Major points

Introduction:

Lines 39ff

“While BEC methods are used to reduce or eliminate the technical differences between matched cells, they introduce new errors that accompany dimension reduction and estimation of batch differences. [...]”

BEC may introduce artifacts, I agree. I think that this sentence implies a certainty about additional artifacts from BEC. One may think of over-integration of batches. However, this statement oversimplifies the entire approach of BEC. Also, how the estimation of batch differences is altered remains unclear. A metric for quantifying batch effects will also take into account the artifacts from BEC. I assume that the authors have some ground truth in mind, that can be simulated and defined by a set of parameters. In this scenario, speaking about distortions caused by BEC makes sense, but it is hardly applicable to real-world data.

Next, the authors do not explain what the distortions may cause in the downstream workflow. For one, insufficient BEC may lead to unclear cell type definitions through clustering. A correct cell type definition is essential for accurate DE analysis. If cell types are consistently defined, one can address the posed hypothesis on differences for DE analysis.

I do not think that “covariate models” are “in contrast” to BEC methods for the following reasons:

1. A possible output of BEC is an integrated data space (integrated graph objects or latent spaces) that allows for consistent cell type annotation. Examples are the outputs of harmony, scanorama, scVI, BBKNN or conos. Here one can only use the uncorrected data for DE analysis.
2. As stated in lines 39-40, a corrected gene expression matrix may be insufficiently corrected for batch effects or introduce artifacts or both. Covariate models can in theory bypass this issue.

Line 49: The assumption of a “balanced” design is quite restrictive, and often not the case in the single-cell RNA-sequencing field. I think that the authors should discuss the validity of this assumption, i.e. how often balanced data are available and how one can create them experimentally.

Line 54ff: I do not understand the point that DE analysis of single-cell RNA-seq data is better than that of bulk RNA-seq data. The reason why DE analysis on single-cell RNA-seq data is more accurate than DE

analysis of bulk RNA-seq data is that one has access to purified populations. On the other hand, single-cell data are more sparse, which limits the power for DE analysis. Pseudobulk approaches could restore the power by aggregating counts in the purified cell populations, however, we then require more samples. The authors should make clear how their study impacts the results on analysing the single-cell data set. Otherwise, the statement that bulk RNA-seq is outperformed by single-cell RNA-seq is not new. Please close the introduction with a teaser to the actual results and be clearer on your findings instead of stating that you performed pathway analysis etc. To be honest: If I were an average reader, the statements “supported the relevance of the results” and “we suggested several methods” would not motivate to read on to the actual result section.

Results:

Covariate models means whether to include the batch covariate in the DE test or not.

In the method section, the authors describe a likelihood ratio test. When DESeq2 is used for DE analysis, I think that one should use the provided likelihood ratio test setting (see the example in <http://bioconductor.org/packages/devel/bioc/vignettes/DESeq2/inst/doc/DESeq2.html#likelihood-ratio-test>). By looking up the code, the authors have run a model that takes into account the condition in the “DESeq2” setting and a model that uses both condition and batch covariate in the DESeq2_Cov model. This is a multi-factor design that uses the standard DESeq2 model for DE analysis. I doubt that this yields the same result as running the likelihood ratio test as outlined in the linked tutorial for DESeq2. The authors should revise both code and method section on the covariate models and clarify in the text what they understand as covariate model.

Note: It took me about 45 min to figure out how the authors actually designed their study, which included to read up on the code on github. I think a graphical abstract would help dramatically to understand the study design. Still, the “LogN” and “voom” methods are not explained in the result section.

I suggest to adapt the synthetic benchmark to include pseudobulk methods.

I suggest to include different log foldchange thresholds in the benchmark as most changes in DE genes may be significant, but the mean expression level is low and few cells drive the differential expression results, which renders the DE gene uninformative.

Briefly explain what a F-score and pAUPR mean, and how one should read the results in Figure 1c-d (e.g. “high F0.5 scores correspond to a high recovery of DE genes”).

When four batches were tested, the F-scores and pAUPRs were lower compared to the scenario with two batches. I think that the authors should state that as a result instead of reporting “similar results”. Such results impose questions on study design, i.e. whether one should increase the number of cells and batches in a study to increase the power, or whether including more data reduces the power of a data set as batch effects dominate over the biological signal.

The authors should add simulation with a less dominant batch effects compared to the biological signal. In this way, it is easier to relate the simulation results to the real-world data analyses.

Figure 2: State in caption that you selected alpha cells and T-cells. Fig. 2a implies that we may look at the same cell type.

Figure 2c: The authors state that “integrative strategies rarely improved the DE analysis of uncorrected data”. I would translate this into: “Applying a DE test on batch-corrected data did not yield higher F0.5-scores compared to running it on uncorrected data.” It is not exactly clear what “integrative strategies” are in contrast to “non-integrative strategies”. Again, a clear terminology and a graphical abstract would help my understanding. Also, DESeq2 shows a huge spread in the F-scores while edgeR performed consistently well at all settings for pancreatic alpha cells.

Figure 3:

How is the error ratio defined? Is it Number of inconsistent signs in DE genes / Total Number of DE genes? Please clarify and state in the x-axis label.

I assume that when one uses only uncorrected data, the sign of a DE gene should be consistent across batches, otherwise it would not be considered a significant result. What does the vertical line represent in Figure 3a and 3c?

Lines 140-141: The authors state a more consistent picture for significantly detected DE genes. Please provide a reference to a figure in the text.

Sign changes:

When looking at significant genes, one still does not take into account the log fold change (logFC). I think that the number of sign changes would reduce even further if one introduces a threshold of $\logFC > 0.5$.

State the total number of DE genes that would be expected in all simulation studies and single-cell data sets.

Test for heterogeneous samples:

I assume that the authors look at lung epithelial cells. Please clarify in text as the original publication sampled from various sites.

The reference to supplementary figure 1 is unfortunate because supplementary figures 2 and 3 have already discussed and I cannot relate the figure description to this paragraph. I suggest that the authors create a separate supplementary figure for this paragraph.

The setup to compare DE genes in LUAD seems arbitrary and messy. I have no intuition of what one should expect as outcome from such a test. Please explain how many common DE genes are observed or expected in this setting. It would be helpful to have a brief description on how an accurate DE result should look like.

Detection of known disease genes:

Lines 226-230: I do not exactly share the authors’ enthusiasm on the “single-cell” outperforms “bulk” finding as this has been clearly shown in a number of single-cell publications and reviews, as I would

consider this common sense (see below).

Detection of prognostic genes:

I think that the comparison to microarray data has several caveats. The microarray data were created from tumour samples of LUAC, so this is a mixture of cells. The resulting gene signature consequently represent a mixture of cell types. Moreover, the most abundant cell types will dominate the signal and cell type composition may vary drastically across patients. Therefore, some genes may be prognostic of the patient's survival, but they did not compare to a null model with a random gene signature.

Pathway analysis seems to be decoupled from the DE analysis benchmark as it focusses on the bulk data vs single-cell RNA-seq data comparison. I would be interested to see how different DE analysis results affect the downstream gene set enrichment analysis.

Discussion:

The discussion largely summarised the results of the paper, and does not clearly give recommendations. Also, I miss a theoretical interpretation of the obtained results. DE analysis is a modelling-heavy field after all and one can clearly address whether some model assumptions hold true or are being violated, which in turn explains certain outcomes.

The last paragraph of the discussion lacks clarity and needs revision. For instance, "For a relevant cell type, many integrative scRNA-seq DE methods" [...] leaves unclear which cell type has been tested and which methods improved the performance of DE methods. The same is the case of the very last sentence.

Methods:

The authors must provide a separate sub-section in the methods section to describe the publicly available datasets they used and how they were pre-processed, including all normalisation methods (e.g. some applications of limma use log-scaled CPM normalised data, which adjusts already for the number of reads/UMIs per cell). The methods section in the supplementary material shows that, for instance, Seurat and scMerge require highly variable genes as input, such that the resulting corrected data matrix has a different number of features than data corrected by MNN or Combat. Figure 7 lists the performance on several datasets that have not been clearly described in the methods section (MCA B cells, pancreas = alpha cells?).

Information and consistency on the pre-processing of the single-cell RNA-seq data is largely missing. Methods like the Seurat Wilcoxon rank sum test operate on scaled data. The authors should extend their study also on the use of scaled data for DE analysis.

Minor points

Introduction:

Line 53: Please clarify that the LUAD data are lung epithelial cells.

Results:

Please state somewhere that “ZW” stands for the Zinb-Wave model, and “modt” for moderated t-test.

Color code in Figure 1c,d for the different methods seems arbitrary.

Figure 4: a,b: y-axis label is misleading. One would rather write “Number of DE genes”, as “Gene counts” refers to the expression level of genes.

Overall: Please use active voice consistently. Use past tense consistently.

Reviewer #3 (Remarks to the Author):

Overall, the idea of performing a comparison of differential gene expression algorithms for single cell data sets is valuable to the scientific community. However there are a number weaknesses in the paper.

The authors inspect the performance of a number of workflows created from batch correction and differential expression algorithms. This is an interesting area to explore, as batch correction approaches often forego count space correction in the interest of efficiency. A number of simulation scenarios are evaluated, along with some actual biological data guided by expected genes.

Referring to 41 methods in the abstract is misleading, as only eight differential expression algorithms are used. It would be more accurate to describe these as workflows, and outline their constituents.

The strength of the study when compared to a more standard DE evaluation (like Squair et al, 2021) is the combined application of batch correction and differential expression. As such, newer batch correction approaches capable of correcting the count space should be included in the benchmark. scVI, scanorama and RISC (<https://www.nature.com/articles/s41587-021-00859-x>) should be present.

Some justification of choosing to go with pAUPR over AUPR should be included.

FDR correction should be more openly acknowledged, and explicitly listed per method.

It would be preferable to have the biological datasets comprehensively evaluated rather than subset to particular cell types for comparison.

The batch specific/cross batch differential expression is not explained well within the results, coming right after a LUAD discussion that served as inspiration. It should be explicitly stated that this is a simulation, especially as the manuscript shifted to using biological data in prior sections.

While encouraging in their findings, the single cell vs. bulk comparisons do not fit the overarching narrative.

The prognostic genes would make more sense to include if an integrative analysis was performed with clinical outcome metadata for the patients.

Given the found impact of sparsity, it would be good to discuss evaluating sparsity in the reader's data as an element to guide workflow selection.

Figure 7A seems to be incorrect, as some numbered fields are coloured "other" rather than "top 5"

The accompanying code repository is somewhat hard to navigate, and the provided description PDF does not match the structure of the folders. Squair et al. created an R package allowing for easy

application of the various DE methods they evaluated. This would be ideal here as well, especially as the workflows can be more complex. At minimum, clear demonstration code examples should be provided for each workflow. `code/simulation_analysis` is the closest to being easily usable for this.

The authors focus on small test datasets, while it's not uncommon for large integrative analyses that would benefit most from the findings of this manuscript to work on millions of cells. Ideally, a data collection of that scale should be used for evaluation purposes, with run time greatly taken into account. At minimum, an explicit recommendation for such scenarios should be made in the discussion, weighting run time and quality of results.

Reviewer #1 (Remarks to the Author):

This manuscript benchmarked different strategies of post-integration differentially gene expression (DEG) analysis with simulated and real single-cell data and recommended parametric DEG methods such as edgeR with batch information as covariate. The manuscript also mentioned that batch effect correction introduces new errors and data distortion and thus DEG analysis on batch corrected gene expression is not recommended.

Comments:

1. It would be helpful to demonstrate the distortion introduced to the high-dimensional gene expression data by the batch correction methods, and whether all batch correction methods introduce such distortion. This will help explain why batch corrected values deteriorate the DEG results.

[Authors' Response] Thank you for this suggestion. For 10 batch-effects-correction (BEC) methods tested in this revision, we compared logFC values of DE genes before and after the correction in Figure 4c and Supplementary Fig. 4c. The logFC values were estimated using log-transformed count data as described in Methods. Some BEC methods increased FCs and some others reduced FCs. Most BEC methods were introducing additional perturbations (noise) to data and expanded or compressed data. These perturbations added artifacts to DE analysis.

2. Simulated DEGs and known disease or prognostic genes were used as ground truth and F1 score and AUC curves were used to quantify the accuracy of DEGs. I wonder whether the magnitude of differential expression (i.e. log fold change between case and control) is taken into consideration.

[Authors' Response] We only used the significance cutoff $q\text{-value} < 0.05$ without considering FC values. For many DE genes in scRNA-seq data, estimation of FC values could be inaccurate because of the noise and missing values. In our test for the sign preservation by DE workflows (Fig. 4), we compared the proportion of simulated DE genes that preserved their signs by each DE workflow. When we restricted the test to only significantly detected DE genes ($q\text{-value} < 0.05$), the error ratios were greatly reduced for all DE workflows (right figures in Fig. 4a,b). When we applied the FC cutoff ($\text{abs}(\log\text{FC}) > 0.5$) in addition to the significance cutoff, a much smaller number of DE genes survived these two cutoffs (Fig. R1), whereas the error ratios for the sign prediction were only slightly reduced. Moreover, the F0.5-scores were considerably lowered by this additional cutoff (Supplementary Fig. 6). Although using both thresholds could be useful for selecting a small number of marker genes, we may not recommend applying logFC cutoff for general DE and function analysis.

Figure R1. The numbers of DE genes survived (left) the cutoff $q\text{-value} < 0.05$ and (right) both the cutoffs $q\text{-value} < 0.05$ and $|\log_2(\text{FC})| > 0.5$ for model-based simulation data.

Do DEG methods influence the genes with large and subtle differences equally or they effect genes with subtle differences more?

[Authors' Response] In the sign prediction test in the section “**Comparison of data distortions in DE analysis**”, the numbers of incorrect sign predictions were dramatically reduced when we applied the significance cutoff $q\text{-value} < 0.05$ (Fig. 4a,b). When we applied an alternative threshold $|\log_2(\text{FC})| > 0.5$, the sign prediction was also greatly reduced (Fig. R2). This implied DE workflows more affected genes with subtle differences.

Figure R2. The proportion of simulated DE genes that satisfied $|\log_2(\text{FC})| > 0.5$ in model-based simulation test.

Likewise, do DEG methods effect highly and lowly/rarely expressed genes equally? A gene that is lowly or rarely expressed should be excluded for DEG analysis even if it has significant p values and large log fold change.

[Authors' Response] We performed additional tests for only top (or bottom) 50% sparsely expressed DEGs with the background (nonDE) genes remaining the same (Fig. R3, below). The precision-recall results for the sparsely expressed DEGs nearly collapsed (Fig. R3, right). This meant most DE workflows failed to detect many sparsely expressed DEGs, and their performances were mainly determined by their ability to detect less sparsely expressed DEGs. This low performance for sparsely expressed genes is one reason why we considered F_{0.5}-score and pAUPR in place of F-score and AUPR. We obtained a very similar results in tests for highly/lowly expressed DE genes. However, we were not able to fully explain some exotic behaviors of edgeR and pseudobulk methods in these tests, which might require further study before publication. We have already excluded very sparsely expressed genes (zero rate > 0.95) in all our simulation and real data analyses. This gene-filtering has a similar effect as excluding lowly expressed genes.

During the revising our manuscript, we found that “sequencing depth” was an important factor that determined the performance of DE workflows. Thus, we added this dimension throughout our work, which required many additional tests. This demonstrated the performance of DE workflows for various combinations of experimental conditions between batch effects, sequencing depth and sparsity, suggesting suitable methods for different situations (Table 1).

Figure R3. Precision-recall for less sparse (50%, left) and more sparse (50%, right) DE genes.

3. Naming of some of the DEG strategies is a bit difficult to follow, for example, ZW could mean the observation weights or corrected gene expression values from Zinb-Wave. I was not sure if raw or corrected values were used for DEG for certain methods.

[Authors' Response] ZINB-WaVE-corrected data were only used for Wilcoxon test, so now we changed the name ZW-Wilcox to ZW_BEC_Wilcox. All other ZW methods used the observation weights. The meaning of ZW was now noted. A similar issue arised with Seurat. Seurat provided BEC data which we used for demonstrating the data distortions in Figure 4c. However, Seurat_Wilcox workflow in this revision did not use Seurat BEC data. Seurat_Wilcox used sctransform normalization incorporating batch covariate in their model and we applied Wilcoxon test to that normalized data. This performed

better than the Wilcoxon test applied to Seurat BEC data in nearly all cases; thus, we included Seurat_Wilcox only.

4. Have the authors tried DEG identification within each batch and compare the results with those from integrative DEGs, then count the frequency of integrative DEGs appearing in individual batch DEG sets. The higher frequency suggests higher confident level for that gene. This is useful when large batch effects are present; no ground-truth DEGs are available; in particular, when the difference between the groups (case vs control) is subtle.

[Authors' Response] Yes, we have compared DE results for individual batches and integrative DE results for LUAD scRNA-seq data in Supplementary Figure S11, where integrative methods overall performed better than most results for individual patients. We think your suggestion to use the frequency in individual batches may help select more reliable DE genes. On the other hand, the DEGs that were not detected in individual batches (often the case with genes with a subtle group difference) can be regarded as a new discovery obtained through "integrative" analysis, so we believe they should not be undervalued.

5. In the section of Preservation of signs of differential expression, I feel that the effects of BEC and DEG are entangled. It would be simpler to first look at log fold change without any DEG methods, and compare log fold change before and after BEC.

[Authors' Response] We appreciate this comment. As you indicated, many DEG methods (except Wilcoxon) use their own methods to estimate fold changes. Therefore, we had only compared signs of DEGs before and after applying each DE workflow instead of comparing logFC values. In this revision, we plotted logFC values before and after BEC without using DE methods. The logFC values were estimated using log-transformed count data (see Methods). This also demonstrated the distortion levels introduced by each BEC method (Figure 4c and Supplementary Fig. 4c)

Reviewer #2 (Remarks to the Author):

Summary

The authors present a benchmarking study on differential expression (DE) analysis methods in single-cell RNAseq and address whether batch-corrected single-cell data can or should be used for DE analysis. The question is timely and addresses the need for accurate DE analysis. While it has been theoretically addressed that DE analysis should not be done on integrated data as they may contain technical artifacts introduced by the batch effect correction (BEC) method, the actual impact has not been systematically addressed. The authors then apply DE analysis to several publicly available single-cell RNAseq datasets and employ a comparison to bulk methods (TCGA and microarray data).

[Authors' Response] We appreciate your comprehensive review which helped substantially improve our manuscript and clarify the main points. Please, check our point-by-point responses below.

Here, the statement that bulk RNA-seq is outperformed by single-cell RNA-seq is not new.

[Authors' Response] To the best of our knowledge, the literature may only show results for a few known marker genes; however, our results presented “statistical significance” (truncated KS test) for the superiority of scRNA-seq DE analysis for the first time.

Overall, DE analysis on BEC data is not more accurate than DE analysis on uncorrected data. Importantly, including the batch covariate in the DE analysis generally improved the accuracy of the results. However, any theoretical explanation on these results is missing.

[Authors' Response] The DE workflows compared in our work used very different approaches. For example, deep learning procedures were often regarded as a “black box”, and it is not easy to elucidate their effects on individual gene’s expression values. In this revision, we demonstrated the data distortions caused by 10 BEC methods by comparing logFC values of DE genes before and after BEC in Figure 4c. All the BEC methods introduced additional perturbations (noise) on the FC values, and some of them systematically increased or decreased the FC values. What we can say is such perturbations of scRNA-seq data did not improve the accuracy of DE analysis for “sparse” data (zero rate > 80%). Interestingly, for less sparse data (zero rate = 40%), many BEC methods considerably improved DE analysis (Supplementary Fig. 7). We also note that incorporating batch covariate neither improve DE analysis for small batch effects. We totally rewrote Discussion including these arguments. We sought additional theoretical aspects of DE analysis for scRNA-seq data by posting questions on the web (e.g. performance of limmatrend vs limmavoom for scRNA-seq data), and it seemed that theoretical ground for the performance of DE methods still remained largely unaddressed for scRNA-seq data.

Also, the authors could substantially contribute to show whether pseudobulk methods may be useful for DE analysis, but the synthetic benchmark does not include pseudobulk methods. I think that the approach of the manuscript is valid, but the work has several shortcomings that limit the power of the results.

[Authors' Response] We now included pseudobulk results in all synthetic and real data benchmark and obtained an interesting result: pseudobulk methods were highly vulnerable to batch effects, so should be avoided for large batch effects.

Major points

Introduction:

Lines 39ff

“While BEC methods are used to reduce or eliminate the technical differences between matched cells,

they introduce new errors that accompany dimension reduction and estimation of batch differences. [...]”

BEC may introduce artifacts, I agree. I think that this sentence implies a certainty about additional artifacts from BEC. One may think of over-integration of batches. However, this statement oversimplifies the entire approach of BEC. Also, how the estimation of batch differences is altered remains unclear. A metric for quantifying batch effects will also take into account the artifacts from BEC. I assume that the authors have some ground truth in mind, that can be simulated and defined by a set of parameters. In this scenario, speaking about distortions caused by BEC makes sense, but it is hardly applicable to real-world data.

[Authors' Response] We agree the sentence could overly simplify the various approaches in BEC algorithms. For example, autoencoder-based methods used both dimension reduction and extension in a nonlinear way. So, as an introduction part, we replaced the sentence with "... they introduce artifacts derived from data transformations or estimation of batch differences". This is a general statement that may not include every different approach.

Next, the authors do not explain what the distortions may cause in the downstream workflow. For one, insufficient BEC may lead to unclear cell type definitions through clustering. A correct cell type definition is essential for accurate DE analysis. If cell types are consistently defined, one can address the posed hypothesis on differences for DE analysis.

[Authors' Response] We appreciate this point. We focused on the effect of distortions on gene-based analysis. In this revision, we demonstrated how each BEC affects FC values of DE genes (Fig. 4c). Most BEC methods added noise to FCs and some BEC methods increased FCs and some others decreased FCs. Whether these perturbations were beneficial or not could only be tested by comparing the performance of DE workflows in identifying DE genes (in terms of F-score, AUPR, false positive control and so on) as well as their signs. Our results indicated most BEC methods did not improve DE analysis for sparse data. We added this point to Discussion.

Clustering and DE analysis are completely separate processes in general. The clustering step only cares about accurate identification of cell classes. A wide range of BEC methods, feature selection, and clustering algorithms are considered for this purpose. Some BEC methods do not yield the corrected expression data, but still can serve to identify the cell classes. Once the cell labels are obtained, all the genes available are reincluded and the optimal integrative DE method is sought. In this latter step, BEC methods that yield batch-corrected data for all genes, covariate model, or meta-analysis can be considered for integrative DE analysis. This explanation was included in the first section of Supplementary Information.

In many cases, cell labels were given *a priori* from clinical information where only the best integrative DE method needs to be sought, and we used the known cell labels (cancer vs. normal for LUAD; severe vs. moderate for COVID-19).

I do not think that "covariate models" are "in contrast" to BEC methods for the following reasons:

1. A possible output of BEC is an integrated data space (integrated graph objects or latent spaces) that allows for consistent cell type annotation. Examples are the outputs of harmony, scanorama, scVI, BBKNN or conos. Here one can only use the uncorrected data for DE analysis.
2. As stated in lines 39-40, a corrected gene expression matrix may be insufficiently corrected for batch effects or introduce artifacts or both. Covariate models can in theory bypass this issue.

[Authors' Response] As you commented, some BEC methods do not yield corrected data (dubbed "BEC data") and only serve to annotate the cells. The corresponding sentences only referred to BEC methods that yield BEC data (now modified as such). Now, we may use "in contrast" because covariate models use uncorrected data. As far as we understand, Scanorama and scVI yield BEC data that can be used for downstream DE analysis. We actually included four additional BEC methods in this revision (scanorama, scVI, scGen and RISC).

Line 49: The assumption of a "balanced" design is quite restrictive, and often not the case in the single-cell RNA-sequencing field. I think that the authors should discuss the validity of this assumption, i.e. how often balanced data are available and how one can create them experimentally.

[Authors' Response] Thank you for your comments. Although it could seem restrictive, the balanced design is essential for batch effects correction in DE analysis. This design has been commonly observed in large-scale single-cell studies where each batch included multiple patients with various group factors, such as disease severity, sex, age, ethnic group, sample preparation methods and cohort region (our newly added COVID-19 example), or in cancer studies where both tumor and nontumor samples were used from the same patients (our lung cancer example). For unbalanced design, we only have to ignore the batch effects in DE analysis. The previous work (Squair et al. 2021) used independent samples without batch information where we have no choice but to ignore batch effects. This explanation was included in Introduction.

Line 54ff: I do not understand the point that DE analysis of single-cell RNA-seq data is better than that of bulk RNA-seq data. The reason why DE analysis on single-cell RNA-seq data is more accurate than DE analysis of bulk RNA-seq data is that one has access to purified populations. On the other hand, single-cell data are more sparse, which limits the power for DE analysis. Pseudobulk approaches could restore the power by aggregating counts in the purified cell populations, however, we then require more samples. The authors should make clear how their study impacts the results on analysing the single-cell data set. Otherwise, the statement that bulk RNA-seq is outperformed by single-cell RNA-seq is not new.

[Authors' Response] To our knowledge, we presented the "statistical tests" for the first time that DE analysis of scRNA-seq data detected important genes better than that of pseudobulk or bulk sample data (please, let us know if this is not true). This significance was also shown in our additional example of COVID-19 data analysis. Previous results may only report some examples for a few well-known marker genes or only conceptually suggest the superiority of scRNA-seq DE analysis. Our work mainly compared various DE methods for scRNA-seq data with multiple batches; however, we also wanted to

show whether scRNA-seq DE analysis across the batches is truly effective for discovering disease-related genes compared to previous approaches that used bulk or pseudobulk data.

The pseudobulk methods were able to address biological variations between replicates; thus, they exhibited good false discovery controls and overall discriminative abilities (Squair et al. 2021). (We note that Squair and colleagues used independent samples that cannot accommodate batch effects into DE analysis.) We showed that pseudobulk methods were vulnerable to batch effects, and DE analyses that considered individual cells as different replicates outperformed the pseudobulk analyses under substantial batch effects.

Please close the introduction with a teaser to the actual results and be clearer on your findings instead of stating that you performed pathway analysis etc. To be honest: If I were an average reader, the statements “supported the relevance of the results” and “we suggested several methods” would not motivate to read on to the actual result section.

[Authors' Response] Thanks for your advice. We simply removed the last two sentences, and included a statement on the test for large-scale data

Results:

Covariate models means whether to include the batch covariate in the DE test or not.

In the method section, the authors describe a likelihood ratio test. When DESeq2 is used for DE analysis, I think that one should use the provided likelihood ratio test setting (see the example in <http://bioconductor.org/packages/devel/bioc/vignettes/DESeq2/inst/doc/DESeq2.html#likelihood-ratio-test>). By looking up the code, the authors have run a model that takes into account the condition in the “DESeq2” setting and a model that uses both condition and batch covariate in the DESeq2_Cov model. This is a multi-factor design that uses the standard DESeq2 model for DE analysis. I doubt that this yields the same result as running the likelihood ratio test as outlined in the linked tutorial for DESeq2. The authors should revise both code and method section on the covariate models and clarify in the text what they understand as covariate model.

Note: It took me about 45 min to figure out how the authors actually designed their study, which included to read up on the code on github. I think a graphical abstract would help dramatically to understand the study design. Still, the “LogN” and “voom” methods are not explained in the result section.

[Authors' Response] We appreciate your time and comments. When we wrote about the likelihood ratio test, we had edgeR procedure in our mind. As you commented, DESeq2 was implemented using the default setting that used Wald test. In addition, limma used voom and moderated *t*-test. “Covariate modeling” in our manuscript included all the DE methods that used the log-count model that included batch covariate, and was described as such in Methods. We now included a graphical overview of our study in Figure 1.

We also included description for “logN” that used log-normalized count data. I found that some methods that used voom merely used the log-transformed data without using the “precision weights”. This has little difference from “logN”; thus, they were removed in this revision. voom was already used in limma and limma_Cov, which were thus renamed as limmavoom and limmavoom_Cov, respectively.

I suggest to adapt the synthetic benchmark to include pseudobulk methods.

[Authors’ Response] Thank you for this comment. We included pseudobulk methods in all our tests. We tested seven batches in place of four to increase the power of pseudobulk methods. Whereas pseudobulk methods were capable of incorporating biological variations between samples, they were highly vulnerable to batch effects.

I suggest to include different log foldchange thresholds in the benchmark as most changes in DE genes may be significant, but the mean expression level is low and few cells drive the differential expression results, which renders the DE gene uninformative.

[Authors’ Response] **[Authors’ Response]** Among the simulated DE genes, less than 50% of them were detected as significant (q -value < 0.05) (Fig. R4, left), and the q -value cutoff alone considerably improved the accuracy of sign predictions by each workflow.

In general, using logFC threshold in addition to the significance cutoff would provide more reliable DE genes. However, for lowly or sparsely expressed genes, the estimation of logFC can be very unstable. The use of offset may help stabilize the analysis, but it causes loss of many lowly expressed DE genes. Moreover, some highly or densely expressed DE genes with small logFC values can also be excluded by the logFC threshold.

We additionally applied the FC threshold $|logFC| > 0.5$ to the significant DE genes (q -value < 0.05). This additional threshold further reduced the number of detected DE genes substantially (Fig. R4, right); however, the corresponding error ratios were only slightly reduced (Supplementary Fig. 5). Moreover, this FC threshold also reduced $F_{0.5}$ -scores (Supplementary Fig. 6). In our opinion, logFC threshold can be useful when selecting a few marker genes for a cell type, but may not be very useful for general DE and functional analysis. Instead, filtering sparsely (or lowly) expressed genes (e.g., zero rate $> 95\%$) in advance would provide reliable DE results.

Figure R4. The proportion of simulated DE genes that survived (left) $q\text{-value} < 0.05$ and (right) both cutoffs $q\text{-value} < 0.05$ & $|\log\text{FC}| > 0.5$.

Briefly explain what a F-score and pAUPR mean, and how one should read the results in Figure 1c-d (e.g. “high F0.5 scores correspond to a high recovery of DE genes”).

[Authors’ Response] We added a motivation of using F0.5 and pAUPR scores in Supplementary Information “A motivation for using pAUPR and $F_{0.5}$ -scores”.

When four batches were tested, the F-scores and pAUPRs were lower compared to the scenario with two batches. I think that the authors should state that as a result instead of reporting “similar results”. Such results impose questions on study design, i.e. whether one should increase the number of cells and batches in a study to increase the power, or whether including more data reduces the power of a data set as batch effects dominate over the biological signal.

[Authors’ Response] In our 4-batch (now, we used seven batches) simulation case, we had to reduce the effect sizes of DE genes in each batch compared to the 2-batch case. Otherwise, the performance of most methods was extremely good because of the increased sample size. In all simulation tests, we controlled the effect sizes so that we can compare the relative performances between methods. We added this explanation to the legend of Supplementary Fig. 1. Therefore, the results between 2-batch and 7-batch should not be compared directly, and we redescribed it as “...yielded similar relative performances between DE workflows”.

The authors should add simulation with a less dominant batch effects compared to the biological signal. In this way, it is easier to relate the simulation results to the real-world data analyses.

[Authors’ Response] Thank you for this comment. We added simulation results for small batch effects in Figure 2 and Supplementary Figure 1.

Figure 2: State in caption that you selected alpha cells and T-cells. Fig. 2a implies that we may look at the same cell type.

[Authors’ Response] The Figure was moved to Supplementary Figure 2 and was revised as suggested.

Figure 2c: The authors state that “integrative strategies rarely improved the DE analysis of uncorrected data”. I would translate this into: “Applying a DE test on batch-corrected data did not yield higher F0.5-scores compared to running it on uncorrected data.” It is not exactly clear what “integrative strategies” are in contrast to “non-integrative strategies”. Again, a clear terminology and a graphical abstract would help my understanding. Also, DESeq2 shows a huge spread in the F-scores while edgeR performed consistently well at all settings for pancreatic alpha cells.

[Authors’ Response] We now added definitions on “integrative strategy” and “naïve DE analysis” in Introduction. A graphical overview of our work was added in Figure 1. The workflows that used DESeq2 often yielded an error when we analyzed sparse data, so we used pseudocount for DESeq2 workflow as recommended in the RNA-seq community. Indeed, in most of our tests, the use of pseudocount

improved and stabilized the results of DESeq2-related workflows. In the previous results, DESeq2 workflows did not use the pseudocount in some instances, which have caused some unstable results. We reanalyzed the whole simulation tests in this revision and the huge spread in DESeq2 was no more observed in the new results.

Figure 3:

How is the error ratio defined? Is it Number of inconsistent signs in DE genes / Total Number of DE genes? Please clarify and state in the x-axis label.

I assume that when one uses only uncorrected data, the sign of a DE gene should be consistent across batches, otherwise it would not be considered a significant result. What does the vertical line represent in Figure 3a and 3c?

[Authors' Response] That is correct. We now clearly defined the “error ratio” in the main text. As you commented, DE genes were simulated so as to have the same signs between batches. The vertical dotted lines indicated the median error ratio for Raw_Wilcox, and we added such explanation.

Lines 140-141: The authors state a more consistent picture for significantly detected DE genes. Please provide a reference to a figure in the text.

[Authors' Response] The section was extended to “Comparison of data distortions in DE analysis” and the Figure numbers were indicated.

Sign changes:

When looking at significant genes, one still does not take into account the log fold change (logFC). I think that the number of sign changes would reduce even further if one introduces a threshold of logFC > 0.5.

[Authors' Response] We tested the logFC threshold ($|\log_2\text{FC}| > 0.5$) in addition to $q\text{-value} < 0.05$ to test the sign consistency of DE genes. Much smaller number of simulated DE genes survived the logFC threshold (Fig. R4); however, the sign error ratios were only slightly reduced (Supplementary Fig. 5). Moreover, this FC threshold reduced $F_{0.5}$ -scores (Supplementary Fig. 6). Thus, logFC threshold could be useful for selecting a small number of marker genes from scRNA-seq data, but may not be recommendable for general DE and function analysis.

State the total number of DE genes that would be expected in all simulation studies and single-cell data sets.

[Authors' Response] In the previous manuscript, we used more than 30% DE genes in the model-based tests. In this revision, we used 20% DE genes (10% up and 10% down) for both model-based and model free tests alike. These proportions are now explicitly stated in the manuscript.

Test for heterogeneous samples:

I assume that the authors look at lung epithelial cells. Please clarify in text as the original publication

sampled from various sites.

The reference to supplementary figure 1 is unfortunate because supplementary figures 2 and 3 have already discussed and I cannot relate the figure description to this paragraph. I suggest that the authors create a separate supplementary figure for this paragraph.

The setup to compare DE genes in LUAD seems arbitrary and messy. I have no intuition of what one should expect as outcome from such a test. Please explain how many common DE genes are observed or expected in this setting. It would be helpful to have a brief description on how an accurate DE result should look like.

[Authors' Response] In this revision, we performed many additional tests, including the tests for additional BEC methods, different sequencing depths and large-scale scRNA-seq data (COVID-19). Therefore, we would remove this relatively less important and intricate section to focus on more important results. We think more analysis would be required to make a clear conclusion about analyzing heterogeneous samples.

Detection of known disease genes:

Lines 226-230: I do not exactly share the authors' enthusiasm on the "single-cell" outperforms "bulk" finding as this has been clearly shown in a number of single-cell publications and reviews, as I would consider this common sense (see below).

[Authors' Response] As far as we understand, previous results may only show results for a few known marker genes; however, our results provide "statistical significance" (truncated KS test) for the first time. Such significance was also observed in our additional analysis for COVID-19 scRNA-seq data. So, we would appreciate it if you could kindly suggest one strong evidence (or example) from the literature that scRNA-seq DE analysis detects important genes better than bulk DE analysis. Then, we could rewrite our findings more appropriately. To our knowledge, previous benchmark papers compared performance of DE methods originally developed for bulk data and those specifically designed for single-cell data using only "single-cell data" (e.g., T. Wang, BMC Bioinformatics 2019), whereas we compared analysis results between bulk and single-cell data. We changed "Remarkably" to "Additionally, we demonstrated that" in the abstract and made a few changes in the text not to overly emphasize this result.

Detection of prognostic genes:

I think that the comparison to microarray data has several caveats. The microarray data were created from tumour samples of LUAC, so this is a mixture of cells. The resulting gene signature consequently represent a mixture of cell types. Moreover, the most abundant cell types will dominate the signal and cell type composition may vary drastically across patients. Therefore, some genes may be prognostic of the patient's survival, but they did not compare to a null model with a random gene signature.

[Authors' Response] We appreciate your insights. (1) Our basic assumption in this test is that the "prognostic" genes exhibit the prognostic signal through a specific cell type. This also implies that expression of that gene in other cell types doesn't have to be correlated with survival. Thus, genes with strong prognostic signal in a cell type can be detected even through bulk sample survival analysis while many others are buried by the noise in unrelated cell types resulting in true negatives.

(2) As you commented, if the proportion of cell types have some prognostic value, then it will be represented by the expression of cell type marker genes in the bulk sample data. Such prognostic genes may not necessarily be detected by DE analysis for a specific cell type with significantly high ranks.

We focused on the first case, and our results demonstrated that many cell type specific prognostic genes were detected from DE analysis of specific cell types (epithelial and myeloid cells) with significantly high ranks.

Pathway analysis seems to be decoupled from the DE analysis benchmark as it focusses on the bulk data vs single-cell RNA-seq data comparison. I would be interested to see how different DE analysis results affect the downstream gene set enrichment analysis.

[Authors' Response] Thank you for your comments. Now, we analyzed the similarity of DE results between 46 workflows by using hierarchical clustering, and selected 10 representative DE workflows for scRNA-seq data from three major clusters to compare their pathway analysis results (preranked GSEA). The GSEA results for the three clusters included some distinct pathways. We also demonstrated the similarity of clustering patterns between LUAD and COVID-19 data.

Discussion:

The discussion largely summarised the results of the paper, and does not clearly give recommendations. Also, I miss a theoretical interpretation of the obtained results. DE analysis is a modelling-heavy field after all and one can clearly address whether some model assumptions hold true or are being violated, which in turn explains certain outcomes.

[Authors' Response] In this revision, we performed many additional tests for different sequencing depths, deep learning techniques, and large-scale scRNA-seq data, and obtained several new interesting results. We totally rewrote Discussion including our interpretation of the results and opinions, and made specific recommendations for different combinations of sparsity, sequencing depths and batch effects (Table 1). We note that the effects of sequencing depth on scRNA-seq DE analysis has been rarely investigated previously. The depth should be taken into account when selecting DE workflow, because shallow sequencing is commonly used recently and many theoretical developments for scRNA-seq data that used zero-inflation model did not fit this low depth data.

The DE workflows compared in our work used very different approaches. In particular, deep learning procedures are often regarded as a “black box”, and it is not easy to elucidate their effects on individual gene's expression values. We demonstrated the data distortions caused by 10 BEC methods by comparing logFC values of DE genes before and after BEC in Figure 4c and Supplementary Figure 4c. Some BEC methods decreased FCs and some others increased FCs, and all the BEC methods introduced additional perturbations (noise) on the FC values. As a third-party comparison, we compared the performance of various DE workflows using known DE genes (simulated data) and known disease

genes (real data), and concluded that the perturbations introduced by BEC methods hardly improved DE analysis for sparse data.

We sought some theoretical aspects by posing questions. We asked how to use BEC data for DE analysis and why limmatrend performed better than limmavoom for scRNA-seq simulation data. The original developer of limma replied that there is no theoretical ground for such difference for single-cell data (so far). The theoretical aspects of scRNA-seq DE analysis still remains largely uninvestigated, especially for low depth data.

The last paragraph of the discussion lacks clarity and needs revision. For instance, “For a relevant cell type, many integrative scRNA-seq DE methods” [...] leaves unclear which cell type has been tested and which methods improved the performance of DE methods. The same is the case of the very last sentence.

[Authors’ Response] We changed “relevant” to “epithelial”. We suggested specific DE workflows for different experimental conditions (Table 1).

Methods:

The authors must provide a separate sub-section in the methods section to describe the publicly available datasets they used and how they were pre-processed, including all normalisation methods (e.g. some applications of limma use log-scaled CPM normalised data, which adjusts already for the number of reads/UMIs per cell). The methods section in the supplementary material shows that, for instance, Seurat and scMerge require highly variable genes as input, such that the resulting corrected data matrix has a different number of features than data corrected by MNN or Combat.

[Authors’ Response] We included a new section, “Acquisition and preprocessing of gene expression data” in Methods and described how genes and cells were filtered, how gene expressed data were acquired and normalized for each workflow, and how we obtained corrected data of the original high dimension in BEC. In Supplementary Information, each DE workflow was explained in much greater detail.

Figure 7 lists the performance on several datasets that have not been clearly described in the methods section (MCA B cells, pancreas = alpha cells?).

[Authors’ Response] MCA B cells were also analyzed before, but the results were excluded because they were similar to T-cell results. The B-cell results were included again in Supplementary Figure 3 in this revision, because this example provided a motivation why we considered F0.5-score and pAUPR. We only analyzed alpha-cells for pancreas data and now we denoted “alpha-cells” together with “pancreatic”.

Information and consistency on the pre-processing of the single-cell RNA-seq data is largely missing. Methods like the Seurat Wilcoxon rank sum test operate on scaled data. The authors should extend their study also on the use of scaled data for DE analysis.

[Authors' Response] We used either raw count data or log-scaled data as input according to the recommended format in each DE workflow. For example, it makes no sense to apply Wilcoxon test for raw count data. Raw_Wilcox used log-scaled data for Wilcoxon test. "Raw" here indicates "uncorrected" data. We note that Tran and colleagues (GB, 2020) also used "Raw" to indicate uncorrected data. How we preprocessed the input data for each DE workflow was described in greater detail in this revision.

Minor points

Introduction:

Line 53: Please clarify that the LUAD data are lung epithelial cells.

[Authors' Response] We analyzed the scRNA-seq data for three major cell types of LUAD, and among them, analysis of LUAD epithelial cells particularly predicted the disease-related genes in significantly high ranks. We added "LUAD" to "epithelial cells".

Results:

Please state somewhere that "ZW" stands for the Zinb-Wave model, and "modt" for moderated t-test.

[Authors' Response] The meaning of ZW was added. Several modt methods were removed in this revision, because modt was already implemented in limmavoom and limmavoom_Cov.

Color code in Figure 1c,d for the different methods seems arbitrary.

[Authors' Response] We used consistent colors for each DE workflow between F0.5-scores and pAUPRs. Only the y-labels of F0.5-scores were colored differently to contrast the performance between different DE categories.

Figure 4: a,b: y-axis label is misleading. One would rather write "Number of DE genes", as "Gene counts" refers to the expression level of genes.

[Authors' Response] It is now Figure 5 and the y-label was changed to "#Genes detected".

Overall: Please use active voice consistently. Use past tense consistently.

[Authors' Response] Thank you for your advice. We revised the writing.

Reviewer #3 (Remarks to the Author):

Overall, the idea of performing a comparison of differential gene expression algorithms for single cell data sets is valuable to the scientific community. However there are a number weaknesses in the paper.

The authors inspect the performance of a number of workflows created from batch correction and differential expression algorithms. This is an interesting area to explore, as batch correction approaches

often forego count space correction in the interest of efficiency. A number of simulation scenarios are evaluated, along with some actual biological data guided by expected genes.

Referring to 41 methods in the abstract is misleading, as only eight differential expression algorithms are used. It would be more accurate to describe these as workflows, and outline their constituents.

[Authors' Response] "41 methods" are now described as 46 workflows.

The strength of the study when compared to a more standard DE evaluation (like Squair et al, 2021) is the combined application of batch correction and differential expression. As such, newer batch correction approaches capable of correcting the count space should be included in the benchmark. scVI, scanorama and RISC should be present.

[Authors' Response] Thank you for the valuable comments. We now included the results for four more BEC methods: scVI, scanorama, RISC and scGen. This even more strengthened our conclusion: The use of BEC data rarely improved DE analysis for sparse data. One exception was limmatrend applied to scVI BEC data outperformed original limmatrend for moderate sequencing depths.

Some justification of choosing to go with pAUPR over AUPR should be included.

[Authors' Response] We added a section on the motivation for using pAUPR and F0.5-score in Supplementary Information.

FDR correction should be more openly acknowledged, and explicitly listed per method.

[Authors' Response] We used Benjamini-Hochberg correction for all DE workflows, and it is acknowledged so.

It would be preferable to have the biological datasets comprehensively evaluated rather than subset to particular cell types for comparison.

[Authors' Response] We used the data for the cell types with top-3 largest proportions for LUAD scRNA-seq data, and performed DE analysis for each cell type comparing cancer and normal cells. Other minor cell types do not have sufficient cell counts for benchmark study, especially for deep learning-based methods. We assume this comment suggests using the cells altogether without discriminating cell types. We think this suggestion could be reasonable because cell type compositions typically change between experimental conditions. However, we think bulk sample analysis is already reflecting this aspect. To understand the mechanisms of cancer development more accurately, it is crucial to compare only cancer and normal epithelial cells excluding immune or blood cells. This selective analysis of the core cell type prioritized known cancer genes significantly better compared to that of other cell types and bulk sample analysis. This shows that analysis of specific cell types is able to reveal gene expression changes in disease in a higher resolution.

The batch specific/cross batch differential expression is not explained well within the results, coming

right after a LUAD discussion that served as inspiration. It should be explicitly stated that this is a simulation, especially as the manuscript shifted to using biological data in prior sections.

[Authors' Response] This comment seems to refer to “**Test for heterogeneous samples**” section. In this revision, we performed many additional tests for more BEC methods, effects of sequencing depths and large-scale scRNA-seq benchmark, and obtained several interesting results. So, we would remove this relatively less important and less clear part to focus on more important results. The best practice for heterogeneous samples may require additional tests to make a clear conclusion.

While encouraging in their findings, the single cell vs. bulk comparisons do not fit the overarching narrative.

[Authors' Response] To the best of our knowledge, our results provide “statistical significance” (truncated KS test) for the first time for the excellence of scRNA-seq DE analysis compared to bulk data analysis. Previous works may only suggest superiority of scRNA-seq DE analysis based on only a few well-known marker genes. We changed “Remarkably” to “Additionally, we demonstrated that” in abstract and made a few changes in the text not to overly emphasize our findings.

The prognostic genes would make more sense to include if an integrative analysis was performed with clinical outcome metadata for the patients.

[Authors' Response] In our analysis of scRNA-seq data for seven lung cancer patients, we have used individual patients as different batches. This already reflects the full diversity of patients' meta data in the covariate modeling and further modification may not be required.

Given the found impact of sparsity, it would be good to discuss evaluating sparsity in the reader's data as an element to guide workflow selection.

[Authors' Response] We totally rewrote Discussion and added specific suggestions regarding batch effects size, sequencing depth, as well as sparsity.

Figure 7A seems to be incorrect, as some numbered fields are coloured "other" rather than "top 5"

[Authors' Response] Thank you for your comment. We totally revised Figure 7 including CUP times and similarity analysis results between DE workflows.

The accompanying code repository is somewhat hard to navigate, and the provided description PDF does not match the structure of the folders. Squair et al. created an R package allowing for easy application of the various DE methods they evaluated. This would be ideal here as well, especially as the workflows can be more complex. At minimum, clear demonstration code examples should be provided for each workflow. code/simulation_analysis is the closest to being easily usable for this.

[Authors' Response]

We have reorganized the structure of the code repository for easier navigation and quick reference. Folder '*method-executing-scripts*' provides all execution for each approach named by the following format: *<method-category>_<method-name>*.

<method-category> including '*bec*', '*cov*', '*meta*' and '*de*' indicates a method such as a BEC, covariate, meta-analysis or DE analysis method. *<method-name>* indicates the specific method implemented. Folder '*an-execution-example-on-2-batch-data*' includes a sample processing workflow with the execution step in a specific order for user's reference. Other details can be found in the updated '*how-to-use-the-code.pdf*' file in the repository.

The authors focus on small test datasets, while it's not uncommon for large integrative analyses that would benefit most from the findings of this manuscript to work on millions of cells. Ideally, a data collection of that scale should be used for evaluation purposes, with run time greatly taken into account. At minimum, an explicit recommendation for such scenarios should be made in the discussion, weighting run time and quality of results.

[Authors' Response] We appreciate this important comment. Tests for large-scale data will make our results more practical. We tested the benchmark methods for the large-scale covid-19 dataset (Ren et al. 2021; 1.4 million cells in total). We note that even when millions of cells are produced, researchers typically select specific cell type of interest and a specific subpopulation to focus on the disease of interest when performing DE analysis. We used 100k monocyte cells from the 48 patient samples that provided fresh/frozen PBMC samples and compared between patients with severe and moderate cases. We benchmarked the workflows for detecting the genes with the GO annotation "defense response to virus", where many DE workflows exhibited significantly high ranks of the virus-related genes (Supplementary Fig. 13). The computation times of each workflow for both LUAD and COVID-19 data were shown in Figure 7a.

REVIEWER COMMENTS

Reviewer #1 (Remarks to the Author):

Thanks the authors for the responses. I have no further questions or comments.

Reviewer #2 (Remarks to the Author):

The authors present the revised version of the manuscript "Benchmarking integration of single-cell differential expression". The manuscript improved substantially upon addressing the reviewer's comments. In particular, the consideration of pseudobulk methods and different strengths of batch effects allowed the authors to draw a much more nuanced picture on the impact of batch correction and DE analysis. As such, the study addresses how batch effect correction affects subsequent data analysis not only on DE analysis, but also on the following Gene Set Enrichment Analysis. Some results of this study do not reproduce previous findings (e.g. regarding pseudobulk methods) and were discussed in detail and set into context.

The results are insightful and a timely contribution to the scientific discourse, overall.

I disagree with the authors on two points, where I kindly ask to provide more context on the first one. The second one might be of interest for the authors, but does not need to be addressed in the manuscript.

Addressing the author's response on my point in "Detection of known disease genes":

Line 258ff "Although the superiority of scRNA-seq DE analysis over bulk RNA-seq analysis has been shared between researchers, it has not been systematically analyzed. Here, we presented statistical tests comparing the performance of scRNA-seq and bulk sample DE analyses."

I think that Sonesson and Robinson (2018), which has been cited as reference 28 in the revised manuscript aims to provide a comparison of bulk vs single-cell DE methods on single-cell RNA-seq data. Moreover, the authors do not clearly distinguish scRNA-seq and bulk RNA-seq as experimental methods and DE analysis methods specifically developed for single-cell and bulk RNA-seq data, respectively. As the authors pointed out in their response, they also compared DE methods on bulk and single-cell RNA-

seq data. As also discussed in the response, bulk data suffer from several shortcomings, which confound the results of a DE test. I think that the authors should clarify their manuscript on this point.

In the discussion, the authors explored the sparsity aspect on DE analysis. The discussion improved massively upon revision, and I would like to respond on the point "The theoretical aspects of scRNA-seq DE analysis still remains largely uninvestigated, especially for low depth data."

In fact, Heimberg et al, Cell Systems, 2016, have addressed the issue of sparsity in single-cell RNA-seq data and investigated the correlation of read depth and number of DE genes/GSEA pathways. This study is largely based on non-microfluidics data with high read depth, though. However, it assesses the aspect of different levels of sparsity in the data, and complements the findings of the manuscript.

Minor:

Fig. 4d – typo: angular cosign distance => should be "angular cosine distance"

Both in the figure caption and the axis label and main text

Please correct.

Fig. S7 – panel labels not included in the figure (I assume that (a) are the upper and (b) the lower panels)

Reviewer #3 (Remarks to the Author):

I am happy with the manner in which the authors addressed my comments. The addition of a larger dataset along with more tailored workflow recommendations strengthens the manuscript.

Some minor points related to the introduced content:

- How were the depth (77, 10, 4) and sparsity (80%, 40%) levels decided?
- Using sex as the batch in the covid data feels rather arbitrary. Was there no better piece of metadata reflecting technical separation that could have been used instead?
- It would be good to make more mention of method performance as the disease scenarios are discussed. This is subsequently summarised in Figure 7c, but this sort of information feels pertinent to the flow of the text.

- I'm having trouble fully interpreting Table 1. Why are there so many fewer rows for 40% sparsity? Clearly stating which findings are simulation-only would be good too.

- The companion repository has improved. I was able to pick a subset of a few methods of interest and peruse the code to determine how I should run them. It would be nice if the code could become more flexible and/or get annotated a bit better. For example, the MAST COV function appears to require cell types hard-coded as Group, along with Batch presumably being the batch. The code then performs a rather convoluted operation to pass Batch as latent.vars in the FindMarkers() call. However, the current state is acceptable.

Reviewer #1 (Remarks to the Author):

Thanks the authors for the responses. I have no further questions or comments.

[Authors' Response] Thank you for your careful comments which substantially helped delineating more comprehensive picture of the topic.

Reviewer #2 (Remarks to the Author):

The authors present the revised version of the manuscript "Benchmarking integration of single-cell differential expression". The manuscript improved substantially upon addressing the reviewer's comments. In particular, the consideration of pseudobulk methods and different strengths of batch effects allowed the authors to draw a much more nuanced picture on the impact of batch correction and DE analysis. As such, the study addresses how batch effect correction affects subsequent data analysis not only on DE analysis, but also on the following Gene Set Enrichment Analysis. Some results of this study do not reproduce previous findings (e.g. regarding pseudobulk methods) and were discussed in detail and set into context.

The results are insightful and a timely contribution to the scientific discourse, overall.

[Authors' Response] We appreciate your thoughtful and critical comments which have led us to do more thorough investigation on integrative DE analysis of single-cell data. Our revision would provide more comprehensive picture of the topic as well as improved insights. We appreciate it.

I disagree with the authors on two points, where I kindly ask to provide more context on the first one. The second one might be of interest for the authors, but does not need to be addressed in the manuscript.

Addressing the author's response on my point in "Detection of known disease genes":

Line 258ff "Although the superiority of scRNA-seq DE analysis over bulk RNA-seq analysis has been shared between researchers, it has not been systematically analyzed. Here, we presented statistical tests comparing the performance of scRNA-seq and bulk sample DE analyses."

I think that Sonesson and Robinson (2018), which has been cited as reference 28 in the revised manuscript aims to provide a comparison of bulk vs single-cell DE methods on single-cell RNA-seq data. Moreover, the authors do not clearly distinguish scRNA-seq and bulk RNA-seq as experimental methods and DE analysis methods specifically developed for single-cell and bulk RNA-seq data, respectively. As the authors pointed out in their response, they also compared DE methods on bulk and single-cell RNA-seq data. As also discussed in the response, bulk data suffer from several shortcomings, which confound the results of a DE test. I think that the authors should clarify their manuscript on this point.

[Authors' Response] Thank you for this comment. We agree that the meaning of "scRNA-seq DE analysis" could be ambiguous. We used this phrase several times to indicate "DE analysis of scRNA-

seq data”, and this is now clearly defined in Introduction. The DE methods specifically developed for scRNA-seq data (e.g., MAST) were already denoted as “single-cell-dedicated” methods in our manuscript. We modified the sentence (258ff) to “Although the superiority of DE analysis of scRNA-seq data over that of bulk RNA-seq data has been expected, it has not been systematically analyzed. Here, we presented statistical tests comparing the performance of scRNA-seq and bulk sample DE analyses in detecting disease-related genes.

Soneson and Robinson (2018) compared Bulk RNA-seq tools and single-cell-dedicated tools in DE analysis of “scRNA-seq data”. We also did this comparison for scRNA-seq data with multiple batches. The sentence (258ff) indicated we performed comparison between DE analysis of scRNA-seq data and DE analysis of large-scale bulk sample data, and concluded that the former exhibited statistically superior performance in detecting disease-related genes.

In the discussion, the authors explored the sparsity aspect on DE analysis. The discussion improved massively upon revision, and I would like to respond on the point “The theoretical aspects of scRNA-seq DE analysis still remains largely uninvestigated, especially for low depth data.”

In fact, Heimberg et al, Cell Systems, 2016, have addressed the issue of sparsity in single-cell RNA-seq data and investigated the correlation of read depth and number of DE genes/GSEA pathways. This study is largely based on non-microfluidics data with high read depth, though. However, it assesses the aspect of different levels of sparsity in the data, and complements the findings of the manuscript.

[Authors’ Response] Thank you for suggesting an interesting reference. This reference investigated how much transcriptional information was retained for low-depth data in dimension reduction analysis and GSEA. We will keep this in mind for future study.

Minor:

Fig. 4d – typo: angular cosign distance => should be "angular cosine distance"

Both in the figure caption and the axis label and main text

Please correct.

[Authors’ Response] Thank you. We corrected the typos.

Fig. S7 – panel labels not included in the figure (I assume that (a) are the upper and (b) the lower panels)

[Authors’ Response] Thank you. The labels were added.

Reviewer #3 (Remarks to the Author):

I am happy with the manner in which the authors addressed my comments. The addition of a larger dataset along with more tailored workflow recommendations strengthens the manuscript.

[Authors' Response] We appreciate your constructive comments that helped substantially strengthen our paper in several aspects.

Some minor points related to the introduced content:

- How were the depth (77, 10, 4) and sparsity (80%, 40%) levels decided?

[Authors' Response] Depth 77 was the depth obtained using splatter simulation package with the default options. Depth 4 was tested because many recently used real scRNA-seq data exhibited depth 4 or lower. We thought the test for depth 10 (rather than $(77+4)/2$) would provide some useful insights for analyzing low depth data.

- Using sex as the batch in the covid data feels rather arbitrary. Was there no better piece of metadata reflecting technical separation that could have been used instead?

[Authors' Response] The purpose of this test was to use large-scale data, and “sex” variable allowed the largest extension. We also considered using technical variables such as fresh/frozen samples and 3' vs 5' sequencing platforms. However, only 0.8% and 0.2% of mild samples used frozen and 3' platform, respectively (unbalanced). How to integrate very large-scale data including multiple biological and technical covariates may require further study.

- It would be good to make more mention of method performance as the disease scenarios are discussed. This is subsequently summarised in Figure 7c, but this sort of information feels pertinent to the flow of the text.

[Authors' Response] We briefly described the performance of workflows in the section “**A gross performance comparison**”

- I'm having trouble fully interpreting Table 1. Why are there so many fewer rows for 40% sparsity? Clearly stating which findings are simulation-only would be good too.

[Authors' Response] By adding the “depth” dimension in our revision, the number of test cases was greatly increased. For 40% zero rates, some results seemed similar to previously known bulk data analysis results. For example, the use of several BEC methods improved DE analysis; however, we did not expect some new findings for small batch effects for this sparsity. We wanted to focus more on “single-cell-specific” results (80% sparsity) instead of analyzing every gradual change and subtle difference between test cases.

The 40% sparsity case included recommended methods based on simulation results only. For 40% sparsity and small batch effects, Most DE methods in Table 1 without BEC are recommendable. These explanations are now included in the legend of Table 1. For the sparsity of 80%, we used simulation results as well as pAUC and tKS p -values of LUAD and COVID-19 scRNA-seq data analyses to select recommended methods, because LUAD and COVID-19 scRNA-seq data exhibited a sparsity of approximately 80%.

- The companion repository has improved. I was able to pick a subset of a few methods of interest and peruse the code to determine how I should run them. It would be nice if the code could become more flexible and/or get annotated a bit better. For example, the MAST COV function appears to require cell types hard-coded as Group, along with Batch presumably being the batch. The code then performs a rather convoluted operation to pass Batch as latent.vars in the FindMarkers() call. However, the current state is acceptable.

[Authors' Response] Thank you for accepting our revision of the code repository. We agree that your suggestion will help use our code easier; however, we find it would be laborious and hard to re-custom our whole repository again without any further problems. For the moment, we would opt to preserve the current state for stability and apply the suggested changes in the next version of the repository.

REVIEWERS' COMMENTS

Reviewer #2 (Remarks to the Author):

All major and minor points have been fully addressed.

Reviewer #3 (Remarks to the Author):

The reviewers have largely addressed the comments that we have made. However, it would be nice for them to include the depth of explanation from the response in the main text for readers to see as well.